


# COSMO-CLM Regional Climate Simulations in the CORDEX framework: a review

Silje Lund Sørland[1,*], Roman Brogli[1], Praveen Kumar Pothapakula[2], Emmanuele Russo[3], Jonas Van de Walle[4], Bodo Ahrens[2], Ivonne Anders[5,6], Edoardo Bucchignani[7,8], Edouard L. Davin[1], Marie-Estelle Demory[1], Alessandro Dosio[9], Hendrik Feldmann[10], Barbara Früh[11], Beate Geyer[12], Klaus Keuler[13], Donghyun Lee[14], Delei Li[15], Nicole P.M. van Lipzig[4], Seung-Ki Min[14], Hans-Jürgen Paniz[10], Burkhardt Rockel[12], Christoph Schär[1], Christian Steger[11], and Wim Thiery[16]

[1]Institute for Atmospheric and Climate Science, ETH Zurich, Switzerland
[2]Institute for Atmospheric and Environmental Sciences, Goethe University Frankfurt am Main, Germany
[3]Climate and Environmental Physics, University of Bern, Switzerland
[4]Department of Earth and Environmental Sciences, KU Leuven, Belgium
[5]Deutsches Klimarechenzentrum, Germany
[6]Central Institute for Meteorology and Geodynamics (ZAMG), Vienna, Austria
[7]Centro Italiano Ricerche Aerospaziali (CIRA), Capua, Italy
[8]Centro Euro-Mediterraneo sui Cambiamenti Climatici (CMCC) Caserta, Italy
[9]European Commission Joint Research Centre (JRC), Ispra, Italy
[10]Institute for Meteorology and Climate Research (IMK-TRO), Karlsruhe Institute of Technology (KIT), Germany
[11]Deutscher Wetterdienst (DWD), 63067 Offenbach, Germany
[12]Helmholtz-Zentrum Geesthacht, Germany
[13]Chair of Atmospheric Processes, Brandenburg University of Technology (BTU) Cottbus - Senftenberg, Germany
[14]Division of Environmental Science and Engineering, Pohang University of Science and Technology (POSTECH), South Korea
[15]CAS Key Laboratory of Ocean Circulation and Waves, Institute of Oceanology, Chinese Academy of Sciences, Qingdao, China
[16]Department of Hydrology and Hydraulic Engineering, Vrije Universiteit Brussel, Belgium
[*]Current affiliation NORCE Norwegian Research Centre, Bergen, Norway

**Correspondence:** ssor@norceresearch.no

**Abstract.** In the last decade, the Climate Limited-area Modeling (CLM) Community has contributed to the Coordinated Regional Climate Downscaling Experiment (CORDEX) with an extensive set of regional climate simulations. Using several versions of the COSMO-CLM community model, ERA-Interim reanalysis and eight Global Climate Models from phase 5 of the Coupled Model Intercomparison Project (CMIP5) were dynamically downscaled with horizontal grid spacings of 0.44°

5  (∼50 km), 0.22° (∼25 km) and 0.11° (∼12 km) over the CORDEX domains Europe, South Asia, East Asia, Australasia and Africa. This major effort resulted in 80 regional climate simulations publicly available through the Earth System Grid Federation (ESGF) web portals for use in impact studies and climate scenario assessments. Here we review the production of these simulations and assess their results in terms of mean near-surface temperature and precipitation to aid the future design of the COSMO-CLM model simulations. It is found that a domain-specific parameter tuning is beneficial, while increasing

10  horizontal model resolution (from 50 to 25 or 12 km grid spacing) alone does not always improve the performance of the simulation. Moreover, the COSMO-CLM performance depends on the driving data. This is generally more important than the





dependence on horizontal resolution, model version and configuration. Our results emphasize the importance of performing regional climate projections in a coordinated way, where guidance from both the global (GCM) and regional (RCM) climate modelling communities is needed to increase the reliability of the GCM-RCM modelling chain.

## 1 Introduction

Dynamical downscaling of global climate models (GCMs) with a regional climate model (RCM) is an approach employed to obtain higher spatial and temporal resolved climate information at the regional to local scale (Rummukainen, 2016; Giorgi, 2019; Gutowski et al., 2016; Jacob et al., 2020). This GCM-RCM model chain data is typically used as the basis for impact studies and long-term adaptation planning by impact modelling groups, stakeholders, and national climate assessment reports (Ahrens et al., 2014; Kjellström et al., 2016; Dalelane et al., 2018; Rineau et al., 2019; Sørland et al., 2020; Sterl et al., 2020; Vanderkelen et al., 2020).

GCM simulations are coordinated through international projects such as the Coupled Model Intercomparison Project phase 5 (CMIP5; Taylor et al. 2012), in which the future scenarios, describing emissions, land-use and aerosol changes, are given by representative concentration pathways (RCPs) (IPCC, 2013; Taylor et al., 2012; Moss et al., 2010). The dynamical downscaling of CMIP5 simulations by RCMs has been initiated through the Coordinated Regional Climate Downscaling Experiment (CORDEX; Giorgi et al. 2009). For the regional climate projections, it is desired to capture all the ensembles of opportunities, i.e. the three dimensional simulation matrix consisting of GCM x RCM x RCP (Knutti et al., 2010a; Hawkins and Sutton, 2009). However, if one RCM were to downscale all CMIP5 simulations for the major continental domains, this would result in a tremendous amount of data and computing time consumption. An initiative such as CORDEX is thus necessary for coordinating the production of regional climate projections (Giorgi et al., 2009). Since 2009, when CORDEX was officially designed and endorsed by the World Climate Research Programme (WCRP), regional climate projections have been produced by several modelling groups over 14 different domains covering nearly all mainlands of the globe. Today, Earth System Grid Federation (ESGF) servers contain more than 370 GCM-RCM model chain simulations (http://htmlpreview.github.io/?http://is-enes-data.github.io/CORDEX_status.html, site accessed 24.11.2020), and the number of simulations has increased substantially in recent years. For instance, for Europe, more than 100 GCM-RCM simulations have been produced as part of EURO-CORDEX. Compared to earlier projects such as PRUDENCE (Christensen and Christensen, 2007) and ENSEMBLES (van der Linden and Mitchell, 2009), the number of simulations has increased by more than 400% (Christensen et al., 2019).

The CORDEX experimental design was initially described in Giorgi et al. (2009), where a minimum horizontal grid spacing of around 0.44° (∼50 km) was recommended. However, it was left to the modelling groups within each CORDEX domain to establish a simulation protocol and to coordinate the simulations. Over Europe, groups were encouraged to perform additional simulations at 0.11° (∼12 km) horizontal resolution (Jacob et al., 2020), although Kotlarski et al. (2014) for Europe as well as Panitz et al. (2014) for Africa found no significant added value in the mean fields with an increase in horizontal resolution. However, added value is found for extreme events and over complex terrain when the grid is refined from 0.44° to 0.11° over Europe (Prein et al., 2016; Torma et al., 2015).





The ensemble size of CORDEX simulations varies greatly amongst domains. Europe has the largest ensemble size, while other domains have a limited number of available simulations (Spinoni et al., 2020). The main reason is the restricted resources from the modelling centers to perform model simulations on the respective domains. To overcome this issue, CORDEX has prioritized domains that are particularly vulnerable to climate variability and change, and for which RCM-based climate projections are rare, such as Africa (Giorgi et al., 2009).

A new framework within CORDEX was presented by Gutowski et al. (2016) (Coordinated Output for Regional Evaluations, CORDEX-CORE), with the goal to produce a set of homogeneous high-resolution regional climate projections covering all continents. A core set of three GCMs from CMIP5 was suggested to be dynamically downscaled for two emission scenarios, with a recommended horizontal grid spacing of 0.22° (~25 km), which is half the horizontal resolution considered in the first CORDEX framework (Giorgi et al., 2009). To participate in the CORDEX-CORE initiative, each RCM group needs to produce

more than 6000 model years, which results in over 400 TB of data, as each domain generates 10 model integrations including 1 evaluation (30 years), 3 historical (3x55 years), 3 RCP2.6 (3x95 years) and 3 RCP8.5 (3x95 years) simulations. This is a huge effort that most RCM groups are not able to perform alone, and until today only two groups were able to conduct all required simulations following the CORDEX-CORE protocol (Remedio et al., 2019; Ciarlo et al., 2020; Teichmann et al., 2020).

The regional climate model COSMO-CLM (or CCLM) is an example of a model developed and used by a community of

scientists, the CLM-Community (http://www.clm-community.eu/). The COSMO-CLM model has been used for a large set of experiments and run over a wide range of resolutions (e.g. Ban et al. 2014; Brisson et al. 2015; Chatterjee et al. 2017; Wouters et al. 2017; Leutwyler et al. 2017; Schultze and Rockel 2018; Schlemmer et al. 2018; Imamovic et al. 2019; Panosetti et al. 2019; Hentgen et al. 2019; Brogli et al. 2019). COSMO-CLM has been used to perform regional climate simulations over Europe for more than 15 years (Rockel et al., 2008), and has today been extensively used for climate simulations over

multiple domains around the Globe (e.g. Panitz et al. 2014; Asharaf and Ahrens 2015; Bucchignani et al. 2016b; Keuler et al. 2016; Sørland et al. 2018; Hirsch et al. 2019; Li et al. 2018; Termonia et al. 2018; Di Virgilio et al. 2019; Russo et al. 2020; Drobinski et al. 2020; Evans et al. 2020), and in this way contributed to the CORDEX initiative. Rockel and Geyer (2008) investigated how COSMO-CLM performs over various domains and climate zones when keeping intentionally the same setup as for its "home-domain", which was introduced as model transferability (Takle et al., 2007). One of the main findings was

that the model has difficulties over domains with a climate substantially different from that of Europe, where the RCM has been developed, and the model may need to be re-tuned for specific domains. This re-tuning can for instance be the use of an objective model calibration (Bellprat et al., 2016; Russo et al., 2020), or the use of a different physical parameterization schemes (e.g. convection after Bechtold et al. (2008) instead of Tiedtke (1989), as it was done in CCLM for Australasia) or a higher model top, which is necessary for tropical regions because of the higher tropopause. In CORDEX, COSMO-CLM was

re-tuned for each of the CORDEX domains (see section Section 2.3).

Since the CMIP5 scenario simulations became available, the CLM-Community has downscaled 8 GCMs (see section 3.2). The majority of the dynamical downscaling experiments with COSMO-CLM has been performed following the EURO-CORDEX framework at 0.11° and 0.44° horizontal grid spacings. There are also numerous simulations for other CORDEX domains at 0.44° horizontal resolution, such as Africa (Panitz et al., 2014; Dosio et al., 2015; Dosio and Panitz, 2016), East





Asia (Li et al., 2018, 2020), South Asia (Asharaf and Ahrens, 2015) and Australasia (Di Virgilio et al., 2019; Hirsch et al., 2019). Recently, as part of the CORDEX-CORE initiative, the CLM-Community has contributed with a set of downscaling experiments over Africa, East Asia, Australasia and South Asia, using a horizontal grid spacing of 0.22°. The total number of simulations conducted by the CLM-Community sums up to 80 simulations (Table 1 lists the number of simulations available for each domain with different resolutions and various RCPs).

This study presents the contribution from the CLM-Community to regional climate projections following the directives of the CORDEX framework. Much of the development of COSMO-CLM is done to improve the model performance over Europe, and COSMO-CLM is today realistically simulating the European climate, which is confirmed in different studies (e.g. Kotlarski et al. 2014; Vautard et al. 2020, and Figure 1). That the RCMs tend to have the best performance over their home-domain is noted previously (Takle et al., 2007). Thus, in this study we assess and compare the model performance over Europe with the

four CORDEX-CORE domains Africa, East Asia, Australasia and South Asia. Since the existing COSMO-CLM CORDEX simulations differ in more than one way (i.e. versions, configurations and resolutions), we do not perform a systematic analysis of each simulation but rather a qualified judgement, based on all model integrations that are currently available (as of February 2020). Such an analysis will support the future design of model simulations in the community. The dependence of the model performance on the driving GCM is also discussed.

The following Section 2 gives an overview of the CLM-Community, the model development and a description of the model configurations used for the CORDEX simulations. Section 3 describes the methods and data. The results are presented in Section 4, and we end with a summary and discussion in Section 5.

## 2 CLM-Community and COSMO-CLM model

### 2.1 The CLM-Community and its community effort

The Climate Limited-area Modelling-Community (CLM-Community, https://www.clm-community.eu) is an open, international network of scientists, joining efforts to develop and use community models. For the last 15 years, the community model employed has been COSMO-CLM (Rockel et al., 2008). COSMO-CLM is the climate version of the COSMO model (COnsortium for Small scale MOdelling), a limited-area numerical weather prediction model developed by Deutscher Wetterdienst (DWD) in the 1990s for weather forecasting applications (Steppeler et al., 2003; Baldauf et al., 2011).

The CLM-Community was founded in 2004, and currently includes 212 members from 72 institutions located all over the world (as of November 2020). The aim of the CLM-Community is to coordinate the model development, its evaluation, and to recommend model configurations. Additionally, the community ensures an efficient use of resources with the objective to provide best possible long-term climate simulations and to help answer key questions of climate change at the regional and local scales.

| Domain | ERA-Interim | | MPI-ESM* | | HadGEM** | | CNRM-CM5 | | EC-EARTH | | CanESM2 | | NorESM | | MIROC5 | | Domain sum |
|---|---|---|---|---|---|---|---|---|---|---|---|---|---|---|---|---|---|
| | 0.11/ 0.22 | 0.44 | 0.11/ 0.22 | 0.44 | 0.11/ 0.22 | 0.44 | 0.11/ 0.22 | 0.44 | 0.11/ 0.22 | 0.44 | 0.11/ 0.22 | 0.44 | 0.11/ 0.22 | 0.44 | 0.11/ 0.22 | 0.44 | |
| EUR | 2 | 2 | 6 | 3 | 3 | 1 | 2 | 1 | 4 | 1 | 1 | | | 1 | 2 | 1 | 30 |
| AFR | 2 | 1 | 2 | 2 | 2 | 2 | | 2 | | 2 | | | 2 | | | | 17 |
| AUS | 1 | 1 | 2 | 2 | 2 | | | | | 2 | | | 2 | | | | 12 |
| EAS | 1 | 1 | 2 | 2 | 2 | 2 | | 2 | | 2 | | | | | | | 14 |
| WAS | 1 | | 2 | 1 | | | | | 1 | | | | 2 | | | | 7 |
| GCM sum | 7 | 5 | 14 | 10 | 9 | 5 | 2 | 5 | 5 | 7 | 1 | | 7 | | 2 | 1 | **80** |

**Table 1.** Number of COSMO-CLM simulations available for the different domains (EUR: Europe, AFR: Africa, AUS: Australasia, EAS: East Asia, WAS: South Asia), driven by ERA-Interim (Dee et al., 2011), CanESM2 (Arora et al., 2011; Von Salzen et al., 2013), CNRM-CM5 (Voldoire et al., 2013), EC-EARTH (Hazeleger et al., 2012a, b), HadGEM (HadGEM2-ES (Collins et al., 2011; Martin et al., 2011) and HadGEM-AO (Baek et al., 2013)), MIROC5 (Watanabe et al., 2011), MPI-ESM-LR (Stevens et al., 2013) and NorESM1-M (Iversen et al., 2013). The resp. ERA-Interim simulation is the evaluation run, and is typically from 1979-2010. The GCM driven simulations include a historical simulation (1950-2005) and one or more scenarios RCP2.6/4.5/8.5 (2006-2099). For each domain, up to two different horizontal grid spacings are used: $0.44°$ (50 km), and $0.11°$ (12 km, only for Europe) or $0.22°$ (25 km, for all the other domains). From the GCM's ensembles the first realisation (r1) is used for all the GCMs, except for EC-EARTH (r12), and for MPI-ESM* (three members: r1, r2, and r3). The HadGEM-ES** GCM is used for all domains, except for East Asia, where HadGEM-AO is used.

## 2.2 COSMO-CLM description, developments, and versions

COSMO-CLM is a non-hydrostatic, limited-area atmospheric model designed for applications from the meso-$\beta$ to the meso-$\gamma$ scales (Steppeler et al., 2003). The model describes compressible flow in a moist atmosphere, thereby relying on the primitive thermo-dynamical equations. These equations are solved numerically with a Runge-Kutta time-stepping scheme (Wicker and Skamarock, 2002) on a three-dimensional Arakawa-C grid (Arakawa and Lamb, 1977). This grid is based on rotated geograph-
ical coordinates and a generalized, terrain-following height coordinate (Doms and Baldauf, 2013). The current standard version has 40 non-equidistant vertical levels up to the top boundary of the model domain at 22.7 km, though the number of levels and height top can be changed by the user. At the upper levels, a sponge layer with Rayleigh damping is used, whereby the default model version is damping all the fields against the driving boundary fields above 11 km. Alternative upper level damping can be chosen (e.g., Klemp et al. 2008) as well as the height where the damping occurs. The standard physical parameterizations
include the radiative transfer scheme by Ritter and Geleyn (1992), the Tiedtke parameterization for convection (Tiedtke, 1989), and a turbulent kinetic energy-based surface transfer and planetary boundary layer parameterization (Raschendorfer, 2001). The parameterization of precipitation is based on a four category microphysics scheme that includes cloud water, rain water, snow, and ice (Doms et al., 2013). The soil processes are simulated by the soil-vegetation-atmosphere-transfer sub-model TERRA-ML (Schrodin and Heise, 2002). Here, prognostic equations are solved for soil water content, temperature and ice in





10 soil layers by default. Alternative parameterizations can be employed (e.g., the parameterization of convection by Bechtold et al. (2008), or land-surface models such as VEG3D or the Community Land Model (Will et al., 2017)).

The model versions used for CORDEX-simulations are COSMO-CLM4-8-17 (Panitz et al., 2014; Keuler et al., 2016; Di Virgilio et al., 2019; Hirsch et al., 2019), multiple versions of COSMO-CLM5 (Sørland et al., 2018; Li et al., 2018) and the accelerated version COSMO-crCLIM (Vautard et al., 2020; Pothapakula et al., 2020). The following sections give short descriptions
of the different versions, their main model developments, and new options for different configurations.

**COSMO-CLM4**

Most developments of COSMO-CLM4 were driven by the goal of reducing a cold bias present in the regional climate simulations over Europe. Sensitivity simulations were carried out with different model configurations at a resolution of 0.44° following the ENSEMBLES (van der Linden and Mitchell, 2009) framework over Europe. The main improvements and de-
velopments were related to an introduction of the new RCP scenarios (van Vuuren et al., 2011; Moss et al., 2010), and a new option for a modified albedo treatment adjusting the albedo according to soil moisture between values for dry and saturated soils (Lawrence and Chase, 2007). For the first CORDEX simulations carried out by the CLM-Community (Keuler et al., 2016), the resulting COSMO-CLM4-8-17 version was used. This version was applied over Europe for an ensemble of simulations with horizontal grid spacings of 0.11° (EUR-11) and 0.44° (EUR-44). The same model version was also used over Africa (Panitz
et al., 2014; Dosio et al., 2015; Dosio and Panitz, 2016), South Asia (Asharaf and Ahrens, 2015), and Australasia (Di Virgilio et al., 2019; Hirsch et al., 2019), but with a modified configuration (see section 2.3).

**COSMO-CLM5**

The developments occurring from COSMO-CLM4 to COSMO-CLM5 comprise the possibility to use, besides the standard temporally constant Aerosol Optical Depths (AOD) described in Tanré et al. (1984), two alternative AOD datasets, namely
Tegen (Tegen et al., 1997) and Aerocom (Kinne et al., 2006), which both vary monthly. In addition, the possibility to choose different parameterizations of bare soil evaporation (see e.g. Schulz and Vogel 2020) was also included in COSMO-CLM5. With COSMO-CLM5, a coordinated parameter testing effort together with an objective model calibration (Bellprat et al., 2012) was performed to test new model options and to find a satisfactory model setup for climate simulations over Europe at the 50 km horizontal grid spacing. This led to the recommended model version of COSMO-CLM5-0-6. Most of the latest
CORDEX simulations are performed with COSMO-CLM5, with minor changes that did not influence the model performance significantly from versions 5-0-6 to 5-0-16 (e.g., minor bug-fixes or additional output variables).

**COSMO-crCLIM**

COSMO-crCLIM is an accelerated version of the COSMO model (based on version 4), that has been developed to run on heterogeneous hardware architectures including multicore Central Processing Units (CPUs) and Graphics Processing Units
(GPUs) (Fuhrer et al., 2014; Schär et al., 2020). COSMO-crCLIM was adapted for climate applications (Leutwyler et al.,





2017) and the current configuration includes a new groundwater formulation (Schlemmer et al., 2018). COSMO-crCLIM has been extensively tested over Europe for convection resolving simulations (Leutwyler et al., 2017; Hentgen et al., 2019; Vergara-Temprado et al., 2020). Other adjustments include changing the upper level damping to only relax the vertical velocity instead of all dynamical fields (Klemp et al., 2008). COSMO-crCLIM has been used to produce CORDEX EUR-11 simulations and

has also contributed to CORDEX-CORE simulations over South Asia (WAS-22).

## 2.3 Model configurations and general specifics for CORDEX domains

The CLM-Community coordinates the development of COSMO-CLM and provides a community model with a standard setup, as described in Section 2.2. However, the model configuration can vary depending on the simulation domain and experimental design. For the CORDEX simulations, the model domains and protocols are provided (see www.cordex.org), but some

changes in the model configuration have been applied depending on the domain and resolution to obtain an optimal model performance. Table S1 summarizes the main differences in the configurations of each model version for each domain. The specific decisions made for each model configuration are described in the following sections. In each case, an evaluation run has been performed, where the boundary conditions are taken from the ERA-Interim reanalysis (Dee et al., 2011), resulting in 12 evaluation simulations.

**CORDEX-Europe**

As most of the model development is done to improve European simulation performances, the EUR-11 and EUR-44 CORDEX simulations are performed with the configuration of the model versions described in Section 2.2 and the specific configurations listed in Table S1. At the time of writing, 30 simulations performed with COSMO-CLM exist for the EURO-CORDEX domain, 21 simulations of which performed with the horizontal grid spacing of 0.11° and 9 simulations with 0.44°. These simulations

are forced by either ERA-Interim (Dee et al., 2011) or 7 GCMs under three RCPs (see Table 1 and S2). The results of these simulations have been included in several scientific studies as well as national climate change assessment reports (e.g. Kotlarski et al., 2014; Keuler et al., 2016; Prein et al., 2016; Sørland et al., 2018; Dalelane et al., 2018; Bülow et al., 2019; Shatwell et al., 2019; Sørland et al., 2020; Vanderkelen et al., 2020; Vautard et al., 2020; Demory et al., 2020; Coppola et al., 2020).

**CORDEX-Africa**

The first CORDEX-Africa simulations were performed with a horizontal grid spacing of 0.44° (AFR-44) using COSMO-CLM4-8-17, following the CORDEX-Africa domain configurations (Giorgi et al. 2009; see also Fig 1 in Panitz et al. 2014). 35 vertical levels were used, and to allow the free development of deep convection throughout the whole tropical troposphere, the height of the upper most level was increased from about 23 km to 30 km above sea level. In addition, the bottom height of the Rayleigh-damping layer (Rayleigh, 1877) was increased from its standard value of about 11 km to 18 km. Together,

these settings are referred to as the COSMO-CLM's tropical configuration (Thiery et al., 2015), a configuration used in several subsequent experiments over tropical domains (e.g. Thiery et al., 2016; Brousse et al., 2019; Van de Walle et al., 2019).





Furthermore, the land surface albedo was replaced by a new dataset based on monthly satellite-derived fields for dry and saturated soil (Lawrence and Chase, 2007), which gave more realistic model results over the deserts. Vegetation parameters (Leaf Area Index and Plant Cover) were also prescribed by monthly climatological fields, derived from the ECOCLIMAP
dataset (Masson et al., 2003). These simulations were analyzed by Panitz et al. (2014), Dosio et al. (2015) and Dosio and Panitz (2016), used for climate impact assessments (e.g., Vanderkelen et al., 2018a, b), and compared to the other CORDEX-Africa RCMs in a number of studies (e.g., Dosio et al., 2019, 2020). In Panitz et al. (2014), an additional evaluation simulation at 0.22° was performed to investigate the effect of increasing the horizontal resolution (from 0.44° to 0.22°) and decreasing the time step (from 240 s to 120 s), respectively (see Table S1).

For the next generation CORDEX-CORE simulations over Africa, a horizontal grid spacing of 0.22° (AFR-22) was required. The AFR-44 setup was used as a starting point, but updated with a new model version, COSMO-CLM5-0-15. The number of vertical levels was increased from 35 to 57 to allow for a more detailed representation of the vertical extent. Several tuning parameters were changed according to the findings of Bucchignani et al. (2016a), and two tuning parameters affecting the thickness of the laminar boundary layer for heat (rlam_heat) and the vertical variation of the critical humidity for sub-grid
clouds (uc1) were changed to reduce precipitation and temperature biases. The applied aerosol climatology was also changed from Tanré et al. (1984) to Tegen et al. (1997). At the time of writing, 17 COSMO-CLM CORDEX simulations exist over the African domain (8 for AFR-22 and 9 for AFR-44, see Table 1).

**CORDEX-Australasia**

The Northern part of the CORDEX-Australasia domain extends into the tropics, therefore the tropical setup used over the
CORDEX-Africa domain was employed for the simulation at 0.44° horizontal grid spacing (AUS-44). For convection, the Bechtold scheme (Bechtold et al., 2008) was used instead of the default Tiedtke-scheme (Tiedtke, 1989). For these simulations, CCLM4-8-17 was used, but instead of applying the standard TERRA-scheme (Schrodin and Heise, 2002) mainly developed for mid-latitude climate, CCLM4-8-17 was coupled to the Community Land Model version 3.5 (CLM3.5, Oleson et al., 2008; Davin et al., 2011) to reduce warm biases over the Australian arid areas present in the standard version. The CCLM4-8-17-
CLM3-5 simulations are analyzed in model comparison studies (Di Virgilio et al., 2019; Hirsch et al., 2019) over the Australian part of the CORDEX-Australasia domain.

For the CORDEX-CORE simulations (AUS-22), CCLM-5-0-15 was used, in which a new computation of bare soil evaporation using a resistance formulation was implemented (Schulz and Vogel, 2020). As this implementation substantially improved the near-surface temperature biases, a coupling to CLM3.5 was no longer necessary. 57 vertical levels are employed for the
AUS-22 simulations, otherwise the configuration is identical to the AUS-44 simulations.

For the Australian domain, currently a total of 12 CORDEX simulations exist, of which 7 with the AUS-22 configurations and 5 with the AUS-44 configurations.





**CORDEX-East Asia**

The CORDEX-EAS-44 simulations use CCLM-5-0-2, with 45 vertical levels where the upper most level is at the height of
30 km. A timestep of 300 s is used. Considering the substantial extension of troposphere height across the tropical areas, the
lower boundary of the Rayleigh damping layer in the model was set to 18 km rather than the typical value of 11 km, similar to
the African setup. The tuning parameters are default except for the vertical diffusion coefficient (wichfakt) that was increased.
The standard aerosol dataset was replaced with Tegen (Tegen et al., 1997) aerosol climatology. These simulations have been
applied in scientific studies focusing on model evaluation or projected change in surface temperature, precipitation and wind
speed/energy over CORDEX-EAS (Li et al., 2018, 2019, 2020).

For EAS-22, CCLM-5-0-9 was employed. Compared to CCLM-5-0-2, a minor bug for soil water content transpiration was
fixed. Several namelist parameters are set differently from their default values (Table S1, type of turbulence, microphysics,
convection, and surface schemes). Spectral nudging based on von Storch et al. (2000) was employed to zonal and meridional
winds above 850 hPa to reduce systematic biases in surface air temperature, precipitation and monsoon circulation over East
Asia, while retaining the observed large-scale variations (Lee et al., 2016), supporting previous RCM studies for East Asia
(e.g., Cha et al. 2011; Hong and Chang 2011). A time step of 150 s is used.

14 COSMO-CLM simulations currently exist for the East-Asian domain, of which 5 were performed following the EAS-
22 framework, and 9 following the EAS-44 framework. It should be noted that the CORDEX-East Asia domain has slightly
changed since its initial configuration, thus EAS-22 and EAS-44 cover slightly different domains (Zhou et al., 2016).

**CORDEX-South Asia**

Over South Asia, COSMO-CLM has been tested and used in various downscaling experiments with a horizontal grid spacing
of 0.44° (Rockel and Geyer, 2008; Dobler and Ahrens, 2010, 2011). Yet, the first COSMO-CLM simulation following the
CORDEX protocol for South Asia at 0.44° horizontal grid spacing (WAS-44) was carried out in Asharaf and Ahrens (2015).
A total of 35 vertical levels were used in this configuration with a time step of 240 s. The standard physical schemes were
employed, except for the kessler-type microphysics scheme (Kessler, 1969). The GCM MPI-ESM-LR was downscaled for the
historical and RCP4.5 emission scenarios.

Within the CORDEX-CORE framework, COSMO-crCLIM-v1-1 was used at a horizontal grid spacing of 0.22°, using the
tropical configuration (height top of 30 km) including 57 vertical levels and a time step of 150 s, as suggested by Asharaf and
Ahrens (2015). Except for changes in the horizontal and vertical resolutions, and changes in tuning parameter values based on
expert tuning to improve the model performance, the configuration and parameterization schemes were identical to that over
Europe (see Table S1).

For the South-Asian domain, a total of 6 COSMO-CLM simulations exists following the WAS-22 framework. Note that
for the WAS-44 simulation with CCLM4-8-17, no official evaluation run was performed, thus the downscaled MPI-ESM-LR
(Asharaf and Ahrens, 2015) is only included when analyzing the GCM-driven simulations in section 4.2.



## 3 Method and data

### 3.1 Observational datasets

All simulations are evaluated against a number of global observation datasets, allowing for a fair comparison between the different domains. The main focus is on the performance of near-surface temperature and precipitation. The datasets with their temporal and horizontal resolutions and their references are listed in Table S2.

**Near-Surface Temperature**

Three global near-surface temperature datasets are considered for the evaluation of the simulations. First, the Global Historical Climatology Network version 2 and the Climate Anomaly Monitoring System (GHCN2+CAMS, Fan and van den Dool 2008), which combines two large individual datasets of station observations. Second, a dataset collected by the University of DELaware (UDEL), including a large number of station temperature data, both from the GHCN2 and, more extensively, from the archive of Willmott and Matsuura (2001). Third, time-series datasets produced by the Climatic Research Unit (CRU) at the University of East Anglia, which is based on an archive of monthly mean temperatures provided by more than 4000 weather stations distributed around the world (Jones and Harris, 2008). The three temperature datasets are given as monthly mean and at a horizontal resolution of 0.5° (Table S2).

**Precipitation**

For precipitation, besides the UDEL (Willmott and Matsuura, 2001) and CRU gridded (Jones and Harris, 2008) station data described above, the following datasets are used: the Global Precipitation Climatology Center (GPCC, Schneider et al. 2018), providing monthly gridded precipitation data at 0.25° horizontal grid spacing from quality-controlled weather stations worldwide; the Multi-Source Weighted-Ensemble Precipitation (MSWEP, Beck et al. 2019), including rain gauge, satellite and reanalysis data given at 3-hourly temporal resolution and 0.1° horizontal grid spacing; the Global Precipitation Climatology Project (GPCP, Adler et al. 2003), where data from rain gauge stations, satellites, and sounding observations have been merged to estimate monthly rainfall on a 2.5° global grid; and finally the NOAA Climate Prediction Center (CPC, Chen et al. 2008), providing global daily gauge-based precipitation data on a 0.5° grid.

### 3.2 Model simulation domains, initial and lateral boundary conditions

We present COSMO-CLM simulations performed by the CLM-Community that are following the CORDEX framework (Giorgi et al., 2009; Gutowski et al., 2016), for the domains Europe, Africa, Australasia, East Asia and South Asia. Additional COSMO-CLM simulations have been performed for other CORDEX domains (e.g. Central Asia, Russo et al. (2019, 2020); Antarctica, Zentek and Heinemann (2020); Souverijns et al. (2019); Mediterranean basin, Obermann et al. (2018); South America, Lejeune et al. (2015) and Middle East–North Africa, Bucchignani et al. (2016a, b)). However, as those simulations have not downscaled any of the GCMs used in the current study or are not yet published on an ESGF-node, they are not





considered here. All simulations were carried out in a rotated longitude-latitude spherical coordinate system with grid spacings of 0.11°, 0.22° or 0.44° over the standard CORDEX domains. The simulated COSMO-CLM model domain contains a lateral relaxation zone, which is required by the dynamical downscaling technique to transfer the data of the driving global climate simulation to the regional model simulation.

Soil moisture is initialized by a climatological mean value representative for the starting date of the simulation, taken from an evaluation simulation driven by the ERA-Interim reanalysis (Dee et al., 2011). Following the CORDEX-framework, an evaluation simulation driven by the ERA-Interim reanalysis is performed over each domain, where all the evaluation simulations is covering the time period 1979-2010, except CCLM4-8-17 for EUR-11 and AFR-44 which is simulated for 1989-2008, and AFR-22 CCLM4-8-17 for 1989-2000.

A total of 8 GCMs were downscaled for a continuous transient time period covering the historical period (1950-2005) and the future period (2006-2099) under RCP2.6, RCP4.5 or RCP8.5 (Moss et al., 2010; van Vuuren et al., 2011). Table S3 gives an overview of the simulations performed for each domain, GCM and scenario, similar to Table 1 but including information on the model versions. The GCMs listed below were selected as the driving data for COSMO-CLM simulations because they represent a wide spread of climate changes over Europe, or because they are part of the CORDEX-CORE framework or external projects (e.g. ReKLIS, Dalelane et al. 2018; Vautard et al. 2020):

- CanESM2 (Canada), 210 km (T63), 35 levels (Arora et al., 2011; Von Salzen et al., 2013).

- CNRM-CM5 (France), 160 km (TL127), 31 levels (Voldoire et al., 2013).

- EC-EARTH (Europe), 80 km (T159), 62 levels (Hazeleger et al., 2012a, b).

- HadGEM2-ES (UK), 210 × 140 km, 38 levels (Collins et al., 2011; Martin et al., 2011).

- HadGEM-AO (South Korea), 210 x 140 km (N96), 38 levels (Baek et al., 2013).

- MIROC5 (Japan), 160 km (T85), 40 levels (Watanabe et al., 2011).

- MPI-ESM-LR (Germany), 210 km (T63), 47 levels (Stevens et al., 2013).

- NorESM1-M (Norway) , 270 × 210 km, 26 levels (Iversen et al., 2013).

### 3.3 Evaluation metrics

Near-surface temperature and precipitation are evaluated via the spatial distribution of climatological seasonal means for December-January-February (DJF), March-April-May (MAM), June-July-August (JJA) and September-October-November (SON). The observational datasets are regridded to the CORDEX domains by bilinear and conservative remapping for near-surface temperature and precipitation, respectively. For both variables, biases are calculated as absolute and relative differences between the model and the ensemble mean of the observational products on a grid box level. Accounting for the uncertainty in the observations, the bias is masked (shown in white on maps) when being smaller than the observational range.





To allow an easy comparison of the model performance across domains, we summarize the spatial deviations of the climatological means by Taylor diagrams (Taylor, 2001), which combine the spatial pattern correlation with the ratio of spatial variances. The ensemble mean of the observation datasets is used again as reference. Every data point's distance from the reference corresponds to the normalized and centered root-mean-square difference. The data's standard deviation is normalized relative to the reference, for which the standard deviation is set to 1. For the creation of Taylor diagrams, simulations and

observations were regridded to a common 0.5° grid, and the diagrams were compiled for all land points of the whole regional simulation domain to avoid a subjective area choice for assessing the model performance.

## 4   Results

We focus our discussion on near-surface temperature and precipitation for DJF and JJA, while MAM and SON results are included in the supplementary information. We first describe the reanalysis-driven evaluation runs (analysed for the period of

1981-2010), thereby assessing performance in terms of the importance of model development and configuration versus model resolution for each of the considered CORDEX domains. In the next step, the results of the GCM-driven historical simulations (1981-2010, whereby RCP85 is used for 2006-2010) are analysed, whereby we extend the discussion to include the choice of forcing data.

### 4.1   Evaluation of the reanalysis-driven simulations

As much of the development of COSMO-CLM is done to improve the model performance over Europe, we start by comparing the performance of the evaluation simulations from COSMO-CLM with nine different RCMs that has been developed independently at different European institutions, shown in Figure 1. The COSMO-CLM evaluation simulation is represented by the version COSMO-crCLIM-v1-1. The model performance is assessed in terms of spatial variability over land for the seasonal temperature and precipitation by using a Taylor Diagram (see Section 3.3). It can clearly be seen that the performance of

COSMO-CLM typically lies in the range of the best performing RCMs over Europe. Motivated by this, we then investigate the transferability of the COSMO-CLM model to other domains, and assess the model performance over Europe and compare it to the CORDEX domains Africa, East Asia, Australasia and South Asia.

    Figures 2 and 3 show the near-surface temperature and precipitation biases as derived from the ERA-Interim-driven COSMO-CLM simulations for the five considered domains for JJA and DJF. Table S3 summarizes the mean biases over land for each

evaluation simulation. For reference, the seasonal mean (DJF, MAM, JJA, SON) temperature and precipitation for the different observational datasets is shown in the supplementary information (Figure S1-S10). In the following, a discussion of the characteristic biases for each region is given, seeking to assess if any aspects of the evinced biases in each case could be related to the different model versions or horizontal resolution. Figure 4 is summarizing the model performance for the different domains in a Taylor diagram.



**Figure 1.** Spatial Taylor diagram exploring the model performance of the EUR-11 RCM ensemble, for temperature (upper panels) and precipitation (lower panels) for the boreal summer (June-July-August (JJA); left) and boreal winter (December-January-February (DJF); right) season. The reference observation is the ensemble mean of the products listed in Section 3.1, and the downward facing red triangles indicate every single observational product. The colors represent different ERA-Interim (Dee et al., 2011) driven RCM simulations, whereby the different RCM model versions shown in the legend are: Aladin53, RCA4, RACMO22E, HIRHAM5, REMO2015, WRF331F, HadREM3-GA7-05, RegCM4-6 and CCLM. The latter is represented here by COSMO-crCLIM-v1-1. See Kotlarski et al. (2014) or Vautard et al. (2020) for a documentation and comprehensive comparison of the different RCMs. More details about the evaluation metrics is described in section 3.3.





**Figure 2.** 2-meter air temperature absolute bias ($\mathbf{\Delta_a T_{2m}}$; column 1 and 3) and total seasonal precipitation relative bias ($\mathbf{\Delta_r P}$; column 2 and 4) of the evaluations runs for JJA for the different domains and model resolutions and versions. The evaluation period is from 1981-2010, except EUR-11-CCLM4-8-17 and AFR-44-CCLM4-8-17, which is for the years 1989-2008, and AFR-22-CCLM4-8-17 which is covering the years 1989-2000. The bias is masked white when the model value falls within the observational range. See Table S1 for the model configurations and Table S3 for the full simulation overview.



**Figure 3.** Same as Figure 2, but for DJF.





### 4.1.1 Bias characteristic for the individual domains

**Europe**

The EURO-CORDEX domain covers most of the pan-European region, and thus includes climates characterised by cold continental winters in the northeast, large areas which are influenced by coastal climate, to a dry and warm Mediterranean summer climate. COSMO-CLM has been used to perform regional climate projections over Europe for more than a decade, as part of ENSEMBLES, PRUDENCE, and now EURO-CORDEX projects. In most evaluation studies over Europe, either the E-OBS dataset is used (Kotlarski et al., 2014; Sørland et al., 2018), or the evaluation is performed on higher resolution observations from different countries (Prein et al., 2016). However, here we are using global datasets, to facilitate a fair comparison between the domains. Nevertheless, the bias pattern shown in Figure 2 and 3 for Europe agrees with earlier studies of COSMO-CLM (Kotlarski et al., 2014), with a warm and dry (cold and wet) bias during the summer season over southern/south-eastern (north and north-eastern) Europe. During the winter, there is a pronounced cold and wet bias over the whole of Europe, except in northern parts of Scandinavia. For the winter precipitation bias shown in Figure 3, often the spread between the observation datasets is larger than the magnitude of the bias. These bias patterns are also seen in the majority of the European RCMs (Kotlarski et al., 2014), and have been suggested to be related to using outdated aerosol climatology or improperly parameterized processes (e.g. convection, micorphysics or land-surface processes; Vautard et al., 2013; Davin et al., 2016; Sørland et al., 2020).

Following the EURO-CORDEX framework, COSMO-CLM has contributed with simulations using four different model configurations and resolutions, two EUR-44 simulations (CCLM4-8-17 and CCLM5-0-6) and two EUR-11 simulations (CCLM4-8-17 and COSMO-crCLIM). With this ensemble, we can explore the differences between model versions and horizontal resolutions. For the summer temperature bias, changing the horizontal resolution has very little impact, when comparing the version CCLM4-8-17 between EUR-11 and EUR-44. However, during the winter season, the cold bias is slightly reduced in EUR-11, but a larger warm bias is seen over the northern areas. When comparing the model versions, the newer versions tend to have a colder climate than the older model version, so some of the warm bias is removed (e.g., over Southeast Europe), but this is then enhancing the cold bias elsewhere (e.g., over North-Northeast Europe).

The precipitation bias is similar between the model versions, configurations and resolutions, but there is a tendency for the higher resolution simulations to be wetter, which is reducing the dry bias over e.g. eastern Europe in summer, but then the wet bias is increased, seen over the north-eastern parts.

The mean biases over land for temperature and precipitation (Table S3) suggest that the EUR-44-CCLM4-8-17 simulation tends to have largest mean biases, followed by the two EUR-11 simulations, while the EUR-44-CCLM5-0-6 has lowest mean biases. The fact that the lower resolution simulation with CCLM5-0-6 tends to have a better performance is likely a result of that most of the tuning and testing to improve the model performance with COSMO-CLM over Europe has been done on this particular model version and resolution.





## Africa

As Africa is among the most vulnerable regions to climate change (Giorgi et al., 2009; Niang et al., 2014), and in recent years there has been a huge effort to produce regional climate projections across Africa (e.g. Nikulin et al., 2012; Kothe et al., 2014;
Dosio et al., 2015; Thiery et al., 2016). However, due to the African continent's large and cross-equatorial extent, the CORDEX model domain is covering multiple climatic zones, from southern mid-latitudes over moist tropical to desert climates, yielding a challenge for RCM-modelling groups to set up an optimal model configuration. The COSMO-CLM ensemble over Africa consists of two model versions, the CCLM4-8-17 following the AFR-44 framework, and CCLM5-0-15 for AFR-22 (Table S1). Moreover, as part of the study by Panitz et al. (2014), the CCLM4-8-17 was used to simulate over the African domain
with a higher resolution (AFR-22), mainly to investigate the effect of increased horizontal resolution, while keeping most of the configuration unchanged (only the time step was changed, see Table S1). Thus, with the 3-member CCLM ensemble over Africa, we can investigate the effect of employing different model versions (i.e. AFR-22 CCLM4-8-17 vs. AFR-22 CCLM-5-0-15) and the effect of increased resolution (i.e. AFR-44 vs. AFR-22). The general performance of COSMO-CLM over Africa shows that the summer (winter) hemisphere tends to exhibit a warm (cold) temperature bias (Figure 2-3), which is
assumed to be caused by a wrong representation of clouds, especially at the Intertropical Convergence Zone (ITCZ) (Kothe et al., 2014). The most striking result is that the model performance is very little influenced when using the same model version with almost the same configuration, but different horizontal resolution, consistent with the findings in (Panitz et al., 2014). When the horizontal resolution is increased together with using an updated model version and modifying the configurations, the results for AFR-22 and AFR-44 differ more. Thus, the model performance seems to be more sensitive to model version and
configuration than to the horizontal resolution, and this is seen for both the temperature and precipitation for all the seasons. The AFR-22 simulation with CCLM5-0-15 has been run with an increased number of vertical levels, and changes in the aerosol climatology and some of the tuning parameters compared to the simulations with the older model version. These results suggest that it is not enough to only change the horizontal resolution, but it is important to re-tune the model configuration to the new resolution employed, and similar findings are found when using other RCMs (Wu et al., 2020).
The newer and higher resolution model (AFR-22 CCLM5-0-15) has the lowest model bias in terms of JJA temperature bias, where for instance the warm bias over Sahara is reduced. Nevertheless, the reduction in the warm bias is enhancing the cold bias in the winter season. The JJA precipitation bias is also lower in the AFR-22 CCLM5-0-15 simulation, but the bias-dipole due to poor ITCZ representation is still present. A lower DJF precipitation bias is also observed for the AFR-22 CCLM5-0-15 simulations.

## 400 Australasia

The Australasian CORDEX domain is centered around the mainland Australian continent, covering different climate zones due to the large extent. The northern part has a tropical climate, while the southern part is more sub-tropical with mild winters. While a large part of Australia is categorized as arid or semi-arid regions and this dry surface state is amplifying heat waves (Hirsch et al., 2019), the southern coast and New Zealand have a temperate climate. The COSMO-CLM ensemble over Aus-





tralasia consist of two horizontal resolutions (AUS-22 and AUS-44) with two model versions with quite different configuration, as the AUS-44 CCLM4-8-17-CLM3-5 simulation is coupled to the Community Land Model (Davin et al., 2011), compared to the AUS-22 CCLM-5-0-15 which uses the standard TERRA-ML scheme (Schrodin and Heise, 2002). These differences in both resolution and configuration should be kept in mind when comparing the two sets of simulations. The two evaluation runs exhibit quite different temperature biases, in particular during the austral winter season (i.e. JJA), where the AUS-44 simulation

has a warm bias over most of the Australian continent, compared to a cold bias in the AUS-22 simulation (Figure 2-3). The winter precipitation bias is more similar between the two simulations, with a dry bias over large areas, except over central Australia, which has a wet bias for the AUS-44 simulation. During austral summer (i.e. DJF), a cold temperature bias and dry precipitation bias is seen for both simulations over the tropical regions (i.e. the northern part of the model domain). Elsewhere AUS-44 shows a warm bias, and AUS-22 a warm bias except for the southern coast. The precipitation bias during the summer

resembles the winter pattern, but with larger magnitudes. Based on visual inspection no simulation seems to perform better than the other, and the bias is sometimes within the range of the spread of the observations, in particular for the winter precipitation and summer temperature. However, when comparing mean land biases, the AUS-44 CCLM4-8-17-CLM3-5 simulation exhibits the best performance (Table S4).

**East Asia**

East Asia features high population density, a great variety of topography and vegetation, and complex climate systems, being a vulnerable region to climate change (Konapala et al., 2020). It is strongly influenced by the monsoon system, characterized by a cold dry winter season, with dominant northerly flow from the northern inland, and a warm rainy summer season, with southerly flow advecting moisture from the ocean.

Great efforts have been made to understand the regional monsoon climate over East Asia using regional climate models, 425 starting with the Regional Climate Model Intercomparison Project (RMIP) for Asia (Fu et al., 2005). COSMO-CLM has been used extensively over the region for studying different atmospheric processes, such as surface wind (Feser and von Storch, 2008; Li et al., 2016), as well as the regional climate (Wang et al., 2013; Huang et al., 2015; Zhou et al., 2016; Li et al., 2018).

CORDEX simulations over East Asia at 0.44° (EAS-44) and 0.22° (EAS-22) have been performed with version CCLM5-0-2 and CCLM5-0-9, respectively. Due to an updated EAS-CORDEX domain, the domains are not identical: while the EAS-44 is 430 following the CORDEX framework for the first phase, the EAS-22 is following the second phase (Zhou et al., 2016). Thus, the different domains might have an influence on the model performance. Keeping this effect of the different domains in mind, we compare simulations over East Asia conducted with a similar model version at different horizontal resolutions and with different model configurations.

During boreal summer (Figure 2), EAS-44-CCLM5-0-2 tends to feature a warm bias over East China and part of northwest- 435 ern China and Kazakhstan, while a cold bias is found over southern India and Indochina. In winter (Figure 3), warm biases are widely distributed over the northern part of the East-Asian domain, and large parts of India, while a cold bias is seen over East China, Indochina and the tropical islands. The precipitation during summer shows a dry bias in the same region as with





warm bias, while the wet bias occurs mainly over the Tibetan Plateau. During winter, there is a wet bias of more than 70 % over northern inland, and a dry bias of similar magnitude over India and Indochina.

The EAS-22 simulation shows similar summer bias patterns as EAS-44, including the warm and dry bias in the northwest inland area and the cold bias in the Indochina Peninsula. However, the strong warm and dry bias in EAS-44 over Eastern China is not present in EAS-22. This warm and dry bias in the EAS-44 simulation might be a result of a deficient summer monsoon circulation, where the precipitation over land is not properly simulated. In EAS-22 the bias is reduced, which seems to be due to the use of spectral nudging that is constraining the CCLM-simulation to be closer to the large-scale flow from

ERA-Interim (Lee et al., 2016). In contrast, EAS-22 shows a stronger dry bias over India than seen in EAS-44, which might be associated with the different spatial domains (i.e., larger part of Indian Ocean in EAS-44). During the winter, when the large-scale forcing is stronger, the biases in EAS-44 and EAS-22 are quite similar, suggesting that these biases are related to the physical parameterization schemes used, for instance the deep convection or the land surface scheme. The mean biases over land for the two simulations for the different seasons are of similar magnitude, seen both for temperature and precipitation.

However, it should be noted that the magnitude of the precipitation bias is among the largest of the considered domains (see Table S4), suggesting that the model experiences particular deficiencies in simulating the climate of East Asia.

**South Asia**

The South Asian domain (WAS) comprises several challenging features to simulate properly with a regional model, such as the complex topography from the Himalayan and Hindu-Kush mountain chain in the north, or the tropical climate represented

by a strong seasonal rainfall from the South-Asian monsoon circulation. For the CORDEX WAS domain, only one evaluation integration exists, performed with COSMO-crCLIM-v1-1 at 0.22° grid spacing (WAS-22). During the boreal summer, a cold bias over northern parts of India, the Horn of Africa and Myanmar (Figure 2) is seen. Interestingly, this cold bias is connected with a dry bias as seen over India and parts of the African region. The dry bias over the interior of the Indian subcontinent is also observed in earlier studies where COSMO-CLM is forced with other reanalysis products (e.g., ERA-40 reanalysis in Dobler

and Ahrens, 2010, and NCEP reanalysis II data in Rockel and Geyer, 2008). The dry bias in the summer monsoon rainfall has been attributed to the lack of moisture transport into the interior parts of the Indian subcontinent due to the excess rainfall over the Western Ghats and its nearby warm south-east Arabian sea, and also plausible inconsistencies in the representation of convection (Ahrens et al., 2020). The dry bias is also present in the EAS-22 simulation with its East-Asian domain partly overlapping with the South-Asian domain. Moreover, over the Horn of Africa, the JJA precipitation bias in WAS-22 is similar

to the CCLM biases in the AFR-22 and -44 simulation. Thus, it seems as these biases are not due to the choice of the model configuration or location of the domain, but rather owing to some processes being wrongly represented in COSMO-CLM.

    During the winter season, there is a warm bias over Northwest India and a cold bias over northern Africa and the Middle-East (Figure 3). A similar cold bias is also seen over Africa and the Middle East in the AFR-22 and AFR-44 simulations. For precipitation, a dry bias is seen over most parts of the domain, except for a wet bias in the North-East Himalaya (Figure 3).

This wet bias is also seen in the EAS-22 and EAS-44 simulations.





### 4.1.2 Summarizing the model performance with Taylor diagram

To compare the model performance in terms of spatial variability between the five domains, we explore Taylor diagrams for all the ERA-Interim driven simulations (12 in total). Figure 4 shows the normalized spatial Taylor diagram for precipitation and temperature for the summer and winter seasons. Note that here we use the ensemble mean over all observational datasets, whereas in Figure 2 and 3 the spread between the observations is taken into account. The COSMO-CLM simulations over Europe tend to have the best performance, which is expected since most of the model development for CCLM is done on the European domain, and is consistent with what we would expect from previous discussions regarding model transferability (Takle et al., 2007). When considering the different seasons and variables, it is not evident that increasing the horizontal resolution has a positive impact on model performance. In contrast, a clear improvement can be found for a newer model version, as seen for instance in the precipitation performance for Africa and Europe.

Another element to notice from Figure 4 is that the individual model performance for the simulations for Africa and Europe is not so different, but the same cannot be said for East Asia and Australasia. The model configurations for Africa and Europe only differ in terms of changing the tuning parameters, aerosol climatology, horizontal or vertical resolution (see Table S1). The simulations for Australasia and East Asia differ more in their configurations, resulting in larger differences in the performance score shown by the Taylor diagram, especially seen for the precipitation. The AUS-44 is coupled to the Community Land Model CLM, and this simulations has a better DJF precipitation performance in terms of spatial pattern correlation, but underestimate the spatial variability (see Section 2.3). The configuration used for AUS-22 is closer to the standard COSMO-CLM configuration. Over East Asia, the EAS-22 simulation is using spectral nudging, which is not used in EAS-44, and this seems to also improve performance, in particular for summer monsoon precipitation. Note that the benefit of using spectral nudging has a strong dependency on the forcing data (e.g. Leps et al., 2019).

### 4.2 Evaluation of the GCM driven simulations

The dynamical downscaling of the CMIP5 GCMs provides a great opportunity to produce regional climate projections for the major continental domains globally. While the choice of which GCM to downscale is not trivial, some studies advice on which GCM to prioritize based on the model performance (McSweeney et al., 2015; Jury et al., 2015; Sooraj et al., 2015). In addition, the GCMs used for CORDEX-CORE are chosen based on capturing a large range of the climate sensitivity of the CMIP5 models (https://cordex.org/experiment-guidelines/cordex-core/cordex-core-simulation-framework/).

One of the main benefits of being a community is that the workload can be distributed among, and computing time contributed by, the different groups. A total of eight distinctive GCMs have been dynamically downscaled with COSMO-CLM for five CORDEX domains (see Section 3.2), yielding 80 simulations in total. Even though the size of the model ensemble is varying for each domain, this is an extensive contribution. Figures 5-8 show the model performance for each domain as Taylor diagrams for temperature and precipitation, for the seasons DJF and JJA. All the ERA-Interim evaluation runs are shown in comparison to the GCM driven runs, as is the spread of the observations.





**Figure 4.** Spatial Taylor diagram exploring the model performance for JJA and DJF for precipitation and 2-m air temperature for each domain (labeled with colors) by considering the ERA-Interim driven simulations. The diamonds (circles) are the 12 km or 24 km, respectively (50 km) simulations. The older model version is marked with a white star inside the symbols. The triangle is the mean of all observations.



**Figure 5.** Spatial Taylor diagrams exploring the ERA-Interim and GCM driven simulations for DJF air temperature where only land points are included for the domains Europe, Africa, East Asia, Australasia and South Asia. The colors indicate the forcing data, and the diamond (circle) represents the 12 km or 24 km, respectively (50 km) simulations. The older model version is marked with a small white star within the colored data points. All the different simulations are listed in Table S1. The triangle is the ensemble mean of all the observations, while the upside triangles represent each single observation dataset. The raw ERA-Interim reanalysis is included as a green cross. A zoomed version is shown in Figure S17.



**Figure 6.** Same as Figure 5, but for JJA. A zoomed version is shown in Figure S18.

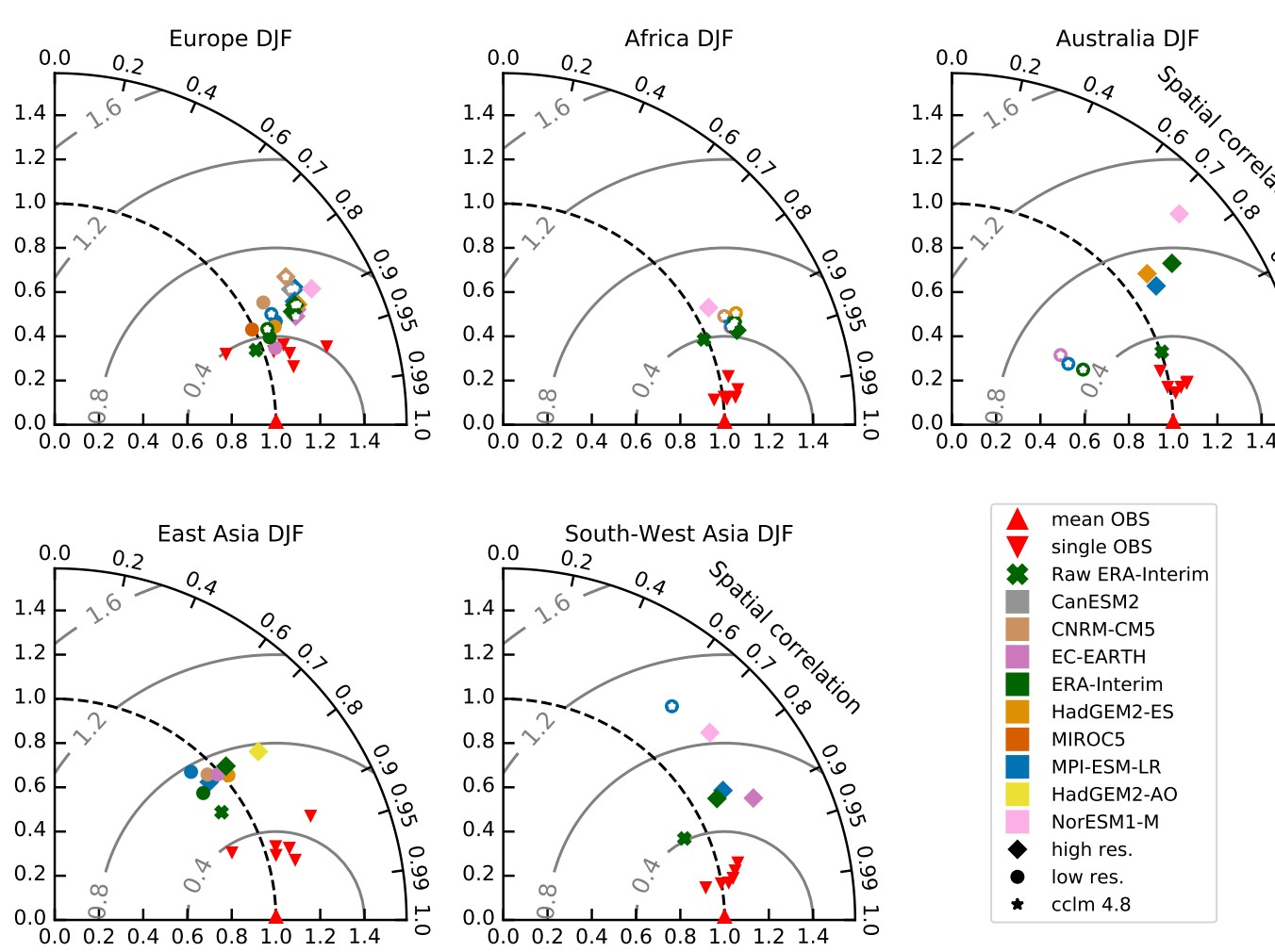

**Figure 7.** Same as Figure 5, but for precipitation. A zoomed version is shown in Figure S19.

**Figure 8.** Same as Figure 7, but for JJA. A zoomed version is shown in Figure S20.





A general result is that the spatial correlation is quite high for the temperature performance, with values larger than 0.9 for most of the domains and simulations. The spatial temperature pattern is dominated by topographical and geographical influence, and tend to be insensitive to the driving GCM, model version and resolution, in particular for DJF, across all domains (Figure 5 and Figure S17). For JJA, higher-resolution simulations are usually closer to the observations (e.g., for Europe, Africa and Australasia, see Figure 6 and Figure S18). The performance of the GCM-driven simulations is typically in the same range as for the ERA-Interim-driven simulations.

For precipitation, the spatial correlation has values down to 0.6 for some simulations, and the spread between the observations is larger for precipitation than for the temperature (Figure 7 and 8). The performance of the individual simulations shows a stronger dependency on the choice of driving data, model version and resolution, which is apparent for both seasons and all domains.

If we consider the individual domains in more detail, we note that for, e.g., Europe, which has the largest model ensemble, the coarser simulations tend to overestimate the temperature spatial variability during winter, while the higher-resolution simulations underestimate it. During summer, almost all model simulations overestimate the spatial variability, and the overestimation is largest for the coarser simulations. This is not a consistent result across the different domains, where for instance for Australasia, the higher resolution simulations have a weak tendency to overestimate the spatial variability for DJF, while this overestimation is lower or even an underestimation of the spatial variability is seen for some of the coarse-resolution simulations.

For precipitation, simulations often overestimate spatial variability. However, as there is a large spread between the observations, with many of the observations having a normalized standard deviation larger than one, it might be that this overestimation is more close to reality. If we consider the individual domains, we see that for Australasia, the two model versions and resolutions have a quite different performance, as also noted in Section 4.1, since the coarser simulation is coupled to CLM. This coupling in the old model version seems to lead to a systematic underestimation of the spatial variance in both seasons independent of the driving model. That the precipitation performance can depend on the driving data, is most clearly seen for Europe, where the downscaled GCM ensemble is largest. However, for the East-Asian domain, when we consider all the GCM-driven simulations, we see that the difference is larger when changing driving data, than when changing model version and configuration, even though the two model versions and their configuration also give different performances, as noticed in Figure 4.

To summarize, for the temperature performance, there is a weak tendency for the higher-resolution simulations to perform better, in particular for JJA, and the choice of driving data has a limited impact on the performance. For precipitation, the choice of the driving data has a bigger influence on the performance compared to the model version and resolution, which is only altering the performance slightly, but not in the same magnitude as when changing the driving data. We see this for all domains, except for Australasia, where the coupling to a different land surface model can be one of the reasons to a change in performance.

If we consider the performance of the simulations that have downscaled the same GCMs, but with different model versions and configurations, for the two horizontal resolutions, there is a tendency for the higher-resolution simulation to show a better





performance (see, e.g., for the downscaled MPI-ESM-LR simulations). This is visible for all domains except for Europe, where the coarser-resolution integrations have the better performance. This can be due to the fact that COSMO-CLM has been 540 developed and targeted to have a good performance over Europe at exactly the coarser resolution. Nevertheless, since not all of the same GCMs have been downscaled for all domains, we cannot make a general conclusion regarding which GCM causes the best performance, and how this performance is depending on the resolution and model version.

## 4.3 Added value of the COSMO-CLM simulations

Until now we have mainly described how the bias patterns and model performance are influenced by changing the model 545 configuration, model version, horizontal resolution, or driving data across the five CORDEX domains. We have shown that an added value in terms of improvements in the model performance is not necessarily gained by only increasing the horizontal resolution, but that the model also has to be re-tuned to obtain a model configuration that is optimal for the domain. Our results also show that investing efforts into model development in terms of improving the physics or adding new features can add value. This is in particular the case for the European domain. Most of the model development has been done on the EUR-44 550 domain, thus the coarser-resolution simulations are outperforming the higher-resolution simulations. However, it should be stressed that we are here only looking into the mean climate, and it has been shown that higher-resolution simulations are adding value when it comes to representing, e.g., the diurnal cycle, the extremes, complex topography or the land-sea contrast (Ban et al., 2014; Torma et al., 2015; Prein et al., 2016; Thiery et al., 2016; Park et al., 2016; Vanden Broucke et al., 2018; Obermann et al., 2018; Helsen et al., 2019; Lee et al., 2020).

Whether an RCM is adding value to the driving GCM data is one of the main motivations to perform dynamical downscaling (Rummukainen, 2016). An RCM inherits its large-scale circulation from the driving data, and any missing information from the boundary conditions is difficult to regenerate by the RCM within the simulation domain (Diaconescu and Laprise, 2013; Hall, 2014; Leps et al., 2019). The ERA-Interim driven simulation is used to evaluate the performance of the RCM, and whether there is an added value over the reanalysis depends on the parameter investigated (e.g., Thiery et al., 2016). Nevertheless, it is 560 hoped that the RCMs should have a similar performance or improve the results of the reanalysis, in particular for the tropical precipitation where reanalysis have poorer skill (Bosilovich et al., 2008). On the other hand, to investigate if an RCM is adding value to the driving GCMs should be done with respect to whether the GCM has a realistic large-scale atmosphere and ocean representation (e.g., Pothapakula et al., 2020).

To assess how the performance of the ERA-Interim driven simulations compares to the skill of ERA-Interim, we have 565 included the reanalysis in the Taylor diagrams, shown in Figures 5 - 8. The spatial pattern of the ERA-Interim bias compared to the different observation datasets is included in the supplementary information (Figure S13-S16). A general result is that the reanalysis is typically closer to the observations than the evaluation simulation. This is not a surprising result, as ERA-Interim is constrained by observations by using a sequential data assimilation scheme (Dee et al., 2011). ERA-Interim agrees well with the spatial variability of the temperature observations, seen mostly for summer, while in winter the reanalysis tends 570 to underestimate the spatial variability (Figure 5 and 6). The temperature in the COSMO-CLM evaluation simulations has a performance similar to the raw ERA-Interim data in terms of spatial pattern correlation, but the CCLM simulations tend





to overestimate the spatial variance. For precipitation, COSMO-CLM has typically a poorer performance than ERA-Interim, seen both for spatial pattern correlation and variability (Figure 7 and 8). In terms of the spatial pattern of the biases from the reanalysis (Figure S13-S16) and CCLM simulations (Figure 2-3), it can be seen that in some areas for the individual

domains, COSMO-CLM has a lower or opposite sign bias than ERA-Interim (e.g., for DJF Africa (southern hemisphere) and India, ERA-Interim has a cold and wet bias, while COSMO-CLM has a warm and dry). However, in most areas ERA-Interim performs better, seen for both temperature and precipitation.

## 5 Summary and outlook

We have presented regional climate simulations performed with the COSMO-CLM following the CORDEX framework (Giorgi

et al., 2009). During this decade of CORDEX, the COSMO-CLM results were influenced by several model upgrades, developments or bug fixes, and model tuning such as parameter testing and objective calibration, and all these advancements had an impact on the model performance. At the same time, as more computing power became available, modelling groups were able to run their model at a higher horizontal resolution, resulting in the CORDEX framework also recommending the RCMs to be run with a horizontal grid spacing of 25 km (12 km for Europe) instead of 50 km which was initially suggested by Giorgi

et al. (2009). When counting the simulations with the distinctive model versions and resolutions, different forcing data and emission scenarios, the CLM-Community has contributed to the CORDEX effort with 80 publicly-available simulations in the ESGF-database spanning five CORDEX domains over the last decade (as of February 2020). This highlights what a comprehensive contribution a community model such as COSMO-CLM can make to the regional climate model ensemble. However, it should be stressed that the COSMO-CLM ensemble is complex and differs in terms of version, configuration, resolution or

driving data, making it challenging to present generic conclusions. Nevertheless, our analysis of all the available model runs, can provide guidance for the future design of regional climate projections by the CLM-Community as well as by other RCM-groups. Moreover, as the focus on downscaling CMIP5 GCMs will be replaced by CMIP6 in the near future, we anticipate this is a good time to reflect how coordinated RCM simulations can contribute in an optimal way. Even though there are increasing research activities aiming at producing continental-scale model ensembles with convection-resolving simulations (Coppola

et al., 2018), or at running global models at a similar resolution as the RCMs (Demory et al., 2020), the use of the dynamical downscaling technique with an RCM at the resolution of 12-25 km will continue to fill an important research need for at least another 5-10 years.

  We have focused on the evaluation simulations (i.e. the ERA-Interim driven simulations) and the GCM-driven simulations in the historical period. One of our main findings is that there is a tendency for higher-resolution simulations to improve model

performance in terms of temperature and precipitation, but much of this improvement is due to model development or model re-tuning to the given domain and resolution, and not only because of better resolved climate processes from an increase in the horizontal resolution. This latter finding is supported by other studies (e.g., Wu et al., 2020). Nevertheless, the positive effect of the higher resolution grid can be disguised as we have only investigated the mean climate, whereas it is expected that a higher resolution will better represent the whole hydrological cycle and extremes (Ban et al., 2014; Torma et al., 2015; Sunyer et al.,





2017; Hentgen et al., 2019). Thus, we emphasize the potential of re-tuning the model for the target domain and horizontal resolution, for example, by increasing the number of vertical levels, by changing the height of the model top, or by performing an objective parameter calibration. Other studies are also suggesting that the convection parameterization could be considered to be switched off at a coarser resolution than what previously thought (Vergara-Temprado et al., 2020). There are additional opportunities to improve model performance by addressing missing or insufficiently represented processes. In particular, using

the most up to date aerosol climatology and including transient aerosol forcing should be considered (Schultze and Rockel, 2018; Gutiérrez et al., 2020; Boé et al., 2020). Similarly, land surface processes representation is an area of regional climate modelling with a lot of room for improvements (Davin et al., 2016). For instance, improving land processes in COSMO-CLM have been shown to positively influence model performance, either through adjustments to the native land surface model in COSMO-CLM (Bellprat et al., 2016; Schlemmer et al., 2018; Akkermans et al., 2012) or by coupling COSMO-CLM to the

Community Land Model (Davin et al., 2011, 2016; Thiery et al., 2015, 2016; Hirsch et al., 2019; Vanden Broucke et al., 2015; Vanden Broucke, 2017). In addition, some specific processes such as the plant physiological response to $CO_2$ increase have been shown to critically influence climate change feedbacks, in particular related to extreme heat (Schwingshackl et al., 2019). The inclusion of land use change forcing is also an area where RCMs lag behind global climate models, despite the recognition that land use impacts are typically stronger at the scales targeted by RCMs and are relevant for decision-making

(Davin et al., 2020). Finally, future RCM developments should consider more explicitly the coupling of the atmospheric model to other components of the climate system, thus transitioning to Regional Earth System Modelling (Giorgi, 2019; Will et al., 2017). An ensemble of regional ocean-atmosphere climate simulations has been performed already within Med-CORDEX for the Mediterranean basin (Somot et al., 2018).

The COSMO-CLM simulations perform better for Europe, and to a lesser extent for Africa, than for the other domains.

As most of the coordinated model development and testing within the CLM-Community has been done to improve the model performance over Europe, this is a confirming and encouraging result. Through different RCM transferability studies, it has been shown that RCMs may respond differently when used over non-native domains, and in particular over regions with contrasting climate (Russo et al., 2020; Takle et al., 2007; Bellprat et al., 2016). Thus, these results suggest that the CLM-Community should improve the coordinated research in the non-European domains, in particular if the goal is to contribute

with dynamical downscaling projections with a global extent. Ideally, coordinated effort should be put into parameter testing for different model resolutions and for new model versions, for all the domains, and not only for Europe.

Another finding is that for the GCM driven simulations, the performance of the simulation has a dependency on the driving data, seen in particular for the precipitation. But, when changing the resolution or slightly altering the model configuration, the performance is only marginally modified, compared to if a substantial adjustment is done in the model configuration (such as

coupling to a different land model as done for AUS-44), this can alter the performance substantially. The results from this large COSMO-CLM model ensemble indicate that an RCM-modeler can do a lot when it comes to improve the model performance, but if there is information missing in the large-scale GCM forcing on the RCM boundaries, it should not be expected that the RCMs can improve on that (Hall, 2014; Pothapakula et al., 2020; Rana et al., 2020). Thus, a coordinated and goal-oriented strategy within CORDEX is needed for selection of the GCM data. Such strategy could address for instance whether only the





GCM performance for each region should be considered, or, whether the spread to include GCM´s sensitivity to increasing greenhouse gasses and other forcings (Rineau et al., 2019) should also be evaluated when selecting driving data. We propose that the planning of the GCM-RCM model chain should be done through coordination between the GCM and RCM-modelers, so that we can obtain a model chain that we trust in and is capturing the range of possible future scenarios (Knutti, 2008).

This paper describes a central and important part of the activities in the CLM-Community within the last decade. COSMO-
CLM was the main workhorse for the contributions of the CLM-Community to CORDEX and to many other projects and activities in the past. Currently, the main developers of the COSMO model, the Deutscher Wetterdienst and its partners in the consortium for small scale modelling, are moving to ICON-based forecasting systems for numerical weather prediction (Zängl et al., 2015). As a consequence, the development of the COSMO model has slowed down over the last years and meanwhile nearly stopped completely. The integration of recent developments and improvements is ongoing as well as the unification of
the numerical weather prediction and CLM-Community branches. COSMO version 6.0 will be released in 2021 and this will be the last official version of the COSMO model.

COSMO-CLM 6.0 will be a state-of-the-art regional climate model and especially the GPU version enables already long-term simulations at convection-resolving resolutions. The model will certainly still be used in several groups of the CLM-Community in the next years. However, the CLM-Community has to prepare for the future. Members of the CLM-Community
have already started to develop a regional climate mode of ICON some years ago in a coordinated effort. A first version of this new regional climate model called ICON-CLM has been prepared in 2019 and a reference simulation has been conducted and analyzed (Pham et al., in review, 2020). The results show that ICON-CLM already performs as good as COSMO-CLM in many aspects and is computationally more efficient. This is very promising, because the model has not been fully optimized for regional climate applications so far, and of course the long-term experience which has been build up in the setup and use
of the COSMO-CLM model is not available yet. This highlights the room for improvements in the near-future. But there are still many technical developments in the model and the infrastructure (mainly pre- and postprocessing) to be done before the modelling system will have the same functionality as COSMO-CLM today.

The transition to ICON will be one of the central topics for the CLM-Community in the next years. Beside this already challenging task, the community will certainly contribute to the downscaling of CMIP6 simulation within the framework of
CORDEX, and possible contributions are currently discussed together with new strategies for the next 5 years. Some of the overarching goals are related to requirements set by new computer architectures, the fact that global climate models will in the next years be able to run at same resolutions as regional models today and possible extensions of the modelling system towards regional Earth System Models that include oceans, dynamic vegetation, a carbon cycle, surface runoff schemes as well as ice sheet and glacier models.

*Code and data availability.* All the official CORDEX simulations used in this study can be downloaded from the ESGF-node: https://esgf-data.dkrz.de/search/cordex-dkrz/.

The WAS-44 simulations are available from: http://cccr.tropmet.res.in/home/ftp_data.jsp.



The YUSPECIF-log files that provide the namelist settings for the different configurations is given as a supplementary file.

The documentation of the COSMO-Model is permanently available:

https://www.dwd.de/EN/ourservices/cosmo_documentation/cosmo_documentation.html.

The COSMO-CLM model is free of charge for all research applications, however, access is license-restricted:

http://www.cosmo-model.org/content/consortium/licencing.htm.

To download the user needs to become a member of the CLM-Community, or the respective institute needs to hold an institutional license.

All observational datasets are publically available:

GHCN-CAMS: https://psl.noaa.gov/data/gridded/data.ghcncams.html

CRU: https://climatedataguide.ucar.edu/climate-data/cru-ts-gridded-precipitation-and-other-meteorological-variables-1901

UDEL: https://psl.noaa.gov/data/gridded/data.UDel_AirT_Precip.html

GPCC: https://www.dwd.de/EN/ourservices/gpcc/gpcc.html,

MSWEP: http://www.gloh2o.org/

GPCP: https://psl.noaa.gov/data/gridded/data.gpcp.html

CPC: https://climatedataguide.ucar.edu/climate-data/cpc-unified-gauge-based-analysis-global-daily-precipitation

*Author contributions.* The COSMO-CLM simulations have been performed by the institutes given in Table S1, which is represented by all the co-authors of the manuscript. The model development has been done as part of community effort. The model simulation data has been collected by Silje Lund Sørland (SLS), and the observation data by Jonas Van de Walle (JVW). The figures have been produced by JVW

and Roman Brogli (RB), with input from SLS, Emmanuele Russo (ER) and Praveen Kumar Pothapakula (PKP), Nicole van Lipzig (NL) and Wim Thiery (WT). The manuscript structure has been prepared by SLS, with input from JVW, RB, ER and PKP. Alessandro Dosio (AD), Burkhardt Rockel (BR), Bodo Ahrens (BA) and NL gave input on a preliminary version of the manuscript. All the co-authors have contributed to the final version of the manuscript.

*Acknowledgements.* The different groups listed in Table S1 that has performed the COSMO-CLM climate simulations acknowledge their

respective supercomputers: ETH simulations at the Swiss Center for Scientific Computing (CSCS, Lugano) using resources from a PRACE allocation. HZG, BTU and DWD simulations using the high performance computing facilities at the German Climate Reseach Centre (DKRZ). Some of the simulations provided by KIT were performed at the HLRS High Performance Computing Center Stuttgart. GUF simulations were performed at the Center for Scientific Computing, Goethe University Frankfurt am Main. The authors acknowledge the World Climate Research Programme's Working Group on Regional Climate for coordinating CORDEX, and the Working Group on Coupled

Modelling responsible panel for CMIP5. The authors also thank the Earth System Grid Federation (ESGF) which host and coordinate the data provision. We also acknowledge NOAA, UCAR, DWD and Princeton University for providing the observational datasets.





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
