# Peer review of "COSMO-CLM Regional Climate Simulations in the CORDEX framework: a review"

_Geoscientific Model Development, 2020_

## Referee Comment (RC2)

A Review of "COSMO-CLM Regional Climate Simulations in the CORDEX framework: a review" by Sørland et al.

General comments

This is a review paper intended to document the development, progress, and performances of COSMO-CLM regional climate simulations over a number of CORDEX domains including Europe, Africa, South Asia, East Asia and Australasia. While there have been many scientific publications of COSMO-CLM over individual domains, this review paper inter-compares its performances among different domains. I find this paper is useful and can provide guidance to future work in regional climate simulations especially in non-native domains of COSMO-CLM. The structure and logical flow of this review paper made it easy to read and follow. I have a few suggestions for further improvement.

Specific comments

1. In Figure 5 to 8, the last Taylor Diagram was labelled as "South-west Asia". I suppose this refers to "South Asia". Perhaps should be changes should be made to supplementary figures as well.

2. "East Asia" was one of the CORDEX domains included in this study. However, this "East Asia" is the "old" domain prior to the establishment of "Southeast Asia" CORDEX domain (e.g. Tangang et al. 2020; www.cordex.org). The new "East Asia" CORDEX domain was shrunk a smaller domain. Hence, a sentence is needed to explain this and avoid confusion.

3. While much have been written on the performances of COSMO-CLM among its different versions and regions, very little review was provided on how this RCM fares compared to other RCMs in different regions.

4. List of GCMs in lines 295 – 303 is better placed in a proper table.

5. I am not sure the real purpose of having a detailed analysis of evaluation of GCM driven simulations in the context of providing a review on COSMO-CLM here? In Figure 5 to 8, the performances of various GCM driven runs were shown to be different. However, these differences are expected and reflect inter-GCM differences. I don't see the relevance of this detailed analysis here in reviewing COSMO-CLM performances.

6. I think in the "Summary and outlook" section, the authors did not adequately address the issue of uncertainty in the simulations. Even within different versions of COSMO-CLM, we can see different biases (e.g. Figure 2 and 3). What does this mean in terms of uncertainty of using COSMO-CLM in different regions? Should this review paper recommend the use of different RCMs i.e. multi-RCM approach?

Ref:
Tangang et al. 2020. Projected Future Changes in Rainfall in Southeast Asia based on CORDEX – SEA Multi-model Simulations. *Climate Dynamics,* **55**, pages1247–1267, https://doi.org/10.1007/s00382-020-05322-2

---

## Author Comment (AC1)

**Response to Community Comment (CC1)  by Jan-Peter Schulz, 03 Mar 2021 (https://doi.org/10.5194/gmd-2020-443-CC1):**

We are very thankful for the comments from Jan-Peter Schulz.  A point-by-point response is given below, where for convenience we have copied in the comments followed by our reply (the J-P.S's comments are in blue text).

A few minor comments on: Soerland et al., 2021: COSMO-CLM Regional Climate Simulations in the CORDEX framework: a review

- Line 124: The reference 'Schrodin and Heise' is slightly wrong. The year is 2001, not 2002. The name of the model is TERRA_LM, not TERRA-ML. Maybe, you say simply TERRA.

Corrected.

- Line 124: Perhaps you could add another reference here, which contains additional characteristics of TERRA which are missing in Schrodin and Heise (2001). This would be Schulz et al. (2016): https://doi.org/10.1127/metz/2016/0537

It also has the advantage that it is a peer-reviewed article, not grey literature.

In total:

"... by the soil-vegetation-atmosphere-transfer sub-model TERRA (Schrodin and Heise, 2001; Schulz et al., 2016). ..."

The suggested reference has been  included in the revised manuscript.

- Line 137: Add the sentence:

"... (Lawrence and Chase, 2007). Furthermore, activating a formulation of soil thermal conductivity dependent on soil moisture was shown to improve the simulated diurnal cycles of the surface temperature, particularly in arid regions (Schulz et al., 2016). For the first CORDEX ..."

The suggested sentence has been  included.

- Line 375: Add reference:

"... Thiery et al., 2016; Schulz et al., 2016). However, ..."

Included.

- Line 1034: Correct reference:

Schrodin and Heise, 2001: TERRA_LM

Corrected.

- Line 1040: Add reference:

Schulz et al., 2016. https://doi.org/10.1127/metz/2016/0537

Included.

---

## Author Comment (AC2)

**Response to Anonymous Referee Comment (RC1), 06 Apr 2021 (https://doi.org/10.5194/gmd-2020-443-RC1):**

We much appreciate this detailed review and the positive assessment provided by the reviewer.

All parts in the manuscript pointed out by the reviewer have been adapted or clarified. A point-by-point response to the comments from the reviewer is given below, where for convenience we have copied in the reviewer's comments followed by our reply (the reviewers' comments are in blue text). When we are referring to a specific line, if nothing else is stated, we are using the line numbering in the document with the track change.

Review of the manuscript entitled "COSMO-CLM Regional Climate Simulations in the CORDEX framework: a review" by S. L. Sorland et al.
This work summarizes the contribution of the CLM community to the CORDEX initiative over several standard domains and nested to different GCMs. Results for near surface temperature and precipitation are provided, compared against several global observation-based data sets. It provides very useful information to understand simulation differences and the authors perform a sound analysis considering multi-model und observational uncertainties. I think it meets the criteria to be published, but some extra effort should be made to further clarify the modelling and analysis details to improve the reproducibility of the results and justify some decisions taken.

I provide next some specific comments:

1) L.26-27 "For the regional climate projections, it is desired to capture all the ensembles of opportunities" Please, rephrase. The use of ensembles of opportunity is not an aim of regional climate projection, but a necessity arising from a lack of a priori design. It is mostly unavoidable to end up producing ensembles of opportunity, but I wouldn't say there is a desire for them.

We have removed this sentence, also as a result of your next comment.

2) L.22-37 Please, consider reorganizing the paragraph. Currently you mention "major continental domains" (L.28) before the CORDEX domains are mentioned a few lines later (L.32). EURO-CORDEX is also first mentioned a few lines later without a reference.

Thanks for pointing this out, we agree that this paragraph was repetitive. We have removed the part where we mention "major continental domains", and also the sentence before (as a result of your previous comment), as we agree that it is not relevant here.

3) L.45-49 This paragraph mixes past and current information. It is quite misleading for the reader. It starts with current qualitative ensemble sizes for different CORDEX domains. Then, it poses Europe as the domain with the currently largest ensemble and, then, refers back to the early days of CORDEX, when the African domain was prioritised, as if this were a recent decision to overcome the imbalance. Moreover,

ensemble size is revisited in this paragraph without adding any new detail with respect to the figures provided e.g. in L.33 or L.35. A dedicated paragraph on domain ensemble sizes is in order but, please, be precise and provide quantitative information.

Thanks again for pointing this out. We have reorganized the paragraph in the following way:

"The ensemble size of CORDEX simulations varies greatly amongst domains. The main reason is the limited resources from the modelling centers to perform model simulations on the respective domains. To overcome this issue, CORDEX has prioritized regions that are particularly vulnerable to climate variability and change, and for which RCM-based climate projections are rare, such as Africa (Giorgi et al. 2009). Still, Europe has the largest ensemble size, while other domains have a smaller number of available simulations (Spinoni et al., 2020).

4) The reader is commonly referred to other publications to obtain basic details. Please, use references as sources of detailed information, but do include the basics in your manuscript. For example: L.57 "until today only two groups [which groups/models?] were able to conduct all required simulations following the CORDEX-CORE protocol" or L.60 "The COSMO-CLM model has been used for a large set of experiments and run over a wide range [what range exactly?] of resolutions"

L58-59: We have now included the information about the RCM-models (i.e. REMO and RegCM)

L63: We have included the information about the resolution range and also added information about the timescales (up to a century)

5) L.92 What do you mean by "qualified judgment"? Please, state clearly the scope of the study.

The challenge with this study is that we cannot assess the effects of the individual model components (version, configuration, resolution and driving data), since the simulations almost always differ in more than one aspect. But that does not mean that we should not provide some 'guidelines for future RCM simulations' or 'lessons learned', where we share our experiences, as we still think we can learn a lot from all these simulations. We have now removed the term "qualified judgment", and replaced it with (on lines 94-97): ", we do not perform a systematic analysis of each simulation, but we rather focus on sharing our experiences, as we anticipate we can learn a lot from this extensive ensemble, which are based on all model integrations that are available as of February 2020. Such an analysis will support the future design of model simulations in the community.."

6) Given that this work tries to present an overview of the CCLM contribution to CORDEX, I think a clearer description of the model genealogy should be provided. Currently, CLM is presented as "Climate Limited-area Modelling", an international network of scientists aiming to develop community models for regional climate research. If I got it right, COSMO-CLM would then be the adaptation of the COSMO

NWP model to climate simulation. Steppeler et al. (2003) is provided as reference for the COSMO model, but this reference does not mention COSMO, but the DWD Lokal Modell (LM). Some renaming seems to have taken place since the early days of CLM in PRUDENCE and ENSEMBLES (L.345). Early references to CLM provide an alternative meaning as the "Climate version of Lokal Modell" (CLM being the model instead of the scientific community) and so does a recent work by Steger and Bucchignani (2020, http://doi.org/10.3390/atmos11111250). Please, better clarify the lineage of the model. Particularly, the specific versions used in PRUDENCE and ENSEMBLES (CLM 2.4.6? according to Jaeger et al, 2008, http://doi.org/10.1127/0941-2948/2008/0301) should be mentioned. The coupling to the CLM (Community Land Model) in Australia adds some extra confusion.

This manuscript is aimed to describe these model versions to avoid confusion for the users, so thanks for pointing this out.  We have included a better description of the model, names and origin on lines 105-116:

*"COSMO-CLM is the climate version of the COSMO (COnsortium for Small scale MOdelling) model (Baldauf et al., 2011), a limited-area numerical weather prediction model developed by Deutscher Wetterdienst (DWD) in the 1990s for weather forecasting applications. COSMO itself is the further developed and renamed version of DWD's "Lokalmodell (LM)" (Steppeler et al., 2003). Based on LM, a Climate version of LM, called CLM, has been developed at the end of the 1990s. In 2007, LM and CLM have been reunified, and, due to the renaming of LM to COSMO, CLM was renamed COSMO-CLM (CCLM: COSMO model in CLimate Mode, see e.g., Rockel et al. 2008; Steger and Bucchignani 2020). The two model branches (COSMO and COSMO-CLM) are developed separately, and merged regularly.  This practice is recognizable in the model version number, where  the whole digit (e.g. 5.0) marks a unified version, and the decimal digit indicates the developments that have occurred independently within the CLM-community and the COSMO-consortium. The new releases  include model improvements, extensions or bug fixes. A new major version is always quality checked and compared to the previous one by means of evaluation of the climatology over the European domain."*

 7) L.151 Is the 5-0-6 or 5-0-16 some kind of semantic versioning system (major-minor-patch)? Some comments in this paragraph seem to imply that model configuration could also be coded in the last number. Can this last nummer be increased because of a particular "recommended" configuration, without any other change in the model? Are the "recommended" versions mentioned in this paragraph the same as the "default" tuning parameters in Table S1?

We have addressed the naming system in our reply above (see also lines 105-116), The main point is that for COSMO-CLM the main version is also the version of the major release (e.g 5.0) and subversions are counted via the "clm" addition (e.g. COSMO-CLM-v5.0_clm14), but when published on ESGF  this was then shortened to e.g. CCLM5-0-14

The recommended version (COSMO-CLM5-0-6) is not using the default tuning parameters, but was objectively calibrated, as stated in Table S1.

 8) L.153 What does crCLIM stand for?

crCLIM is the acronym for a project at ETH (Convection-resolving climate modeling on future supercomputing platforms, http://www.crclim.ch ), which enabled the development of the accelerated version of the COSMO model. See next reply.

 9) L.153-160 Is COSMO-crCLIM still endorsed by the CLM community? From the description given, it apparently branched off CCLM4 and followed an independent development. Will these developments be incorporated back to CCLM6 (this could be added to the outlook paragraphs at the end of the paper)? Is COSMO-crCLIM adopting new CLM developments?

COSMO-crCLIM has been developed in parallel to COSMO-CLM, but there has always been close collaboration to exchange important bug-fixes or developments. All the developments from COSMO-crCLIM will indeed be included in CCLM6, and it is stated on line 698 that "COSMO-CLM 6.0 will be a state-of-the-art regional climate model and especially the GPU version …". However, to make this clearer, we have included a bit more information in the paragraph on lines 167-177, where we explain the acronym, and include information about the merging of the two versions.

10) L.160 Is CORDEX-CORE used as a synonym for simulating at 0.22 deg. resolution? In principle, CORDEX-CORE requires simulating for most domains. It is not just a matter of resolution. Also, in table S1, a CORDEX-CORE framework is stated along with spectral nudging in EAS-22. Would the CCLM CORDEX-CORE contribution mix different model versions (COSMO-crCLIM, COSMO-CLM5-0-x) and nudging settings depending on the domain?

CORDEX-CORE aims at producing a core set of simulations for several (but not all)  CORDEX domains with a target grid resolution in the range of 12.5 -25 km, so it is not simply a synonym for simulating at 0.22 deg. To distinguish between CORDEX and CORDEX-CORE is not straightforward (or fair), as there have been many simulations in the past with a resolution of 50 km for multiple domains (before the CORDEX-CORE framework was established), and we think it is important to also include these when we assess and present CORDEX simulations. Moreover, CORDEX-CORE is very computing and storage demanding, and to participate in such an effort requires a well designed experimental design.  One of the goals of this paper is to share our experience of running CCLM over different domains, using different resolutions, model configurations etc. One of the main take home messages is  that increasing the resolution does not necessarily result in more reliable regional climate predictions. Instead, you can also gain a lot from returning the model configuration to the different domain and also with model development. Since the CORDEX initiative (with downscaling of CMIP5) has been going on for almost a decade, the CLM-Community has continuously done model developments, which has resulted in a contribution from the CLM-Community to CORDEX consisting of various model versions/model configurations. This manuscript is documenting all these differences, with the aim to share the lessons learned. So to answer your last question, yes, the CCLM CORDEX-CORE contribution mixes different model versions and configurations.

In the sentence on line 175  we removed the mention of CORDEX-CORE to avoid confusion. In table S1 we leave the reference to CORDEX-CORE in there, as those simulations have been done with the goal to participate with simulations following the CORDEX-CORE framework.

11) L.182 Was this reduction of the standard vertical levels (40 to 35) done for computational efficiency? It seems odd to increase the top of the atmosphere and reduce the amount of vertical levels.

The major features defining the "African/Tropical setup" are the increases of the bottom height of the Rayleigh-damping layer and the height of the model top. For the CORDEX-Africa (AFR-44) simulations these heights was increased from their standard values for extra-tropical applications (about 11 km bottom height of the Rayleigh-damping layer and 22 km for model top) to 18 km and 30 km, respectively (Panitz et al., 2014). These changes are necessary to allow the free development of deep convection throughout the whole tropical troposphere.

However, for the AFR-44 simulations we had to use 35 vertical levels only due to computing time constraints. We had to apply for computing time at DKRZ and the approved time we got did not allow using more vertical levels. Otherwise it would have not been possible to create the whole CCLM based CORDEX-Africa ensemble (Panitz et al., 2014, Dosio et al., 2015; Dosio et al., 2016). Other options to reduce computing time like reduction of number of horizontal grid-points or increase of numerical time-step were not possible due to the demanded and thus fixed horizontal domain configuration and possible violations of the Courant-Friedrichs-Lewy stability criterion, respectively.

12) L.222 "African setup" was previously defined as "Tropical setup" (L.185). Or does this version include also the other developments for Africa (L.187-190)?

The name tropical setup originates from the African setup, but we have now made sure that we're consistent in the naming and only use "tropical setup" in the manuscript. As described in our previous reply, the major features defining the "African/Tropical setup" are related to the increases of the bottom height of the Rayleigh-damping layer and the height of the model top. Thus, although in other studies (even in other parts of the world, see for example Toelle et al (2017), Lange et al. (2015), Di Virgilio et al., 2019) CCLM simulations used the "Tropical Setup", settings like the number of vertical levels, model tuning parameters and other configuration parameters for the physics (e.g. albedo, aerosol distribution, soil-vegetation model, convection scheme) and dynamics could be and were different from those used in the frame of CORDEX-Africa. We are here documenting these changes in the model configuration for the CORDEX simulations when they have been applied.

see also additional references not cited in the manuscript:

Akkermans, T., Thiery, W., van Lipzig, N.P.M., The regional climate impact of a realistic future deforestation scenario in the Congo Basin, 2014, J. Climate 27(7), 2714-2734.

Docquier, D., Thiery, W., Lhermitte, S., van Lipzig, N.P.M., 2016, Multi-year wind dynamics around Lake Tanganyika, Climate Dynamics, 47(9), 3191-3202.

Lange, St., Rockel, B., Volkholz., J. and Bookhagen, B.: Regional climate model sensitivities to parametrizations of convection and non-precipitating subgrid-scale clouds over South America.
Clim Dyn, 44:2839–2857, DOI 10.1007/s00382-014-2199-0, 2015

Toelle, M., Engler, St and Panitz, H.-J.: Impact of Abrupt Land Cover Changes by Tropical Deforestation on Southeast Asian Climate and Agriculture. Journal of Climate, 30, 2587–2600, 2017

13) L.226 There was only a minor bug fix from version 5-0-2 to 5-0-9? As mentioned above, the reasons to increase this last number in the version specification should be clarified, so the user of the data knows whether different versions can be compared. Apparently, not only model version, but many other subtle changes were applied (Table S1). I think the clear identification of these differences is one of the main outcomes of this paper. With so many small changes, the attribution of the different results to a specific change is problematic, though.

The reviewer is absolutely correct, the main outcome with this paper is to document these changes so that it is easier for the user to know how the various versions differ. The reasoning for the decimal is described now on lines 105-116l (as a result of your comment 6 and 7).

14) Indicate in the caption of Table S3 the meaning of the parenthesis in WAS-44. Is it that no evaluation run is available?

Thanks for pointing this out, and yes that is indeed the meaning. We have now included it in the Table S2 caption.

15) L.252 "allowing for a fair comparison ..." It is not really fair: despite being global, the amount and quality of the background observations used by these data sets greatly differ across domains (e.g. USA or Europe compared to Africa).

Thanks for this comment. It is always a challenge to do a fair comparison over different domains when it comes to observations. We meant fair with respect to the same horizontal resolution and underlying methodology to produce gridded observations, but we see that this can be misleading. We changed the wording to "..here we are using global datasets, in order to compare the model simulations to a common dataset, i.e. with the same horizontal resolution and underlying methodology. ".

16) L.282 How wide is the relaxation zone?

It is between 8-12 grid points (10 x dx). This is added on line 299.

17) L.293 "The GCMs listed below [...] represent a wide spread of climate changes over Europe, or because they are part of the CORDEX-CORE framework or external projects" Which models were selected for each reason? Please, be precise. This GCM selection excluded some CCLM simulations from the study (L.279).

When we started this study, we planned to focus on the domains that were listed as priority domains from the CORDEX-CORE framework (which we state in the

introduction on line 92). We were aware that there were several simulations for other domains, but most of them have only one evaluation simulation, so it would be a challenge to include them in this analysis, and most of the simulations have also not been published on the ESGF-node, which is desired when we evaluate such a large model ensemble for reproducibility. However, we have referred to these other studies extensively in the current manuscript. We feel what we have written on line 291-297 already explains this.

Regarding the selection of GCMs, it is based on what GCMs that have been downscaled within the domains we focus on, as a result of what is described in the paragraph (line 309-313), we rephrase the wording a bit: " *The various GCM used as driving data for COSMO-CLM in this study are listed in Table 2; they include those selected for the CORDEX simulations ( chosen in order to provide a wide range of of climate changes over Europe), and those part of the CORDEX-CORE framework or external projects (e.g. ReKLIS; Dalelane et al. 2018, PRINCIPLES; Vautard et al. 2020).*"

18) L.309 "bias is masked (shown in white on maps) when being smaller than the observational range" Please, clarify the exact procedure for reproducibility purposes. Bias w.r.t. the mean of the observations was masked when the value of the model lies between min(obs) and max(obs)? From the sentence above, it seems that you are computing the observational range R(obs)=max(obs)-min(obs) and masking if abs(bias) < R(obs), i.e: if mean(obs)-R(obs) < model < mean(obs)+R(obs)

We have now added to the sentence to clarify on lines 319 - 322: "Accounting for the uncertainty in the observations, the bias is masked, where white areas indicate areas where model values are within the observational range, which is the minimum and maximum observational values at each grid point."

19) L.331 I would avoid the word "transferability" when model configuration is changed across domains. Transferability experiments are the opposite of your approach: 'perform simulations with all modeling parameters and parameterizations held constant over a specific period on several prescribed domains representing different climatic regions' (Takle et al., 2007). But you use different configurations for each domain and advocate (L.605) for the re-tuning of the model for the target domain.

We have rephrased the sentence to: "*Motivated by this, we then investigate the performance of the COSMO-CLM model over other CORDEX domains, namely Africa, East Asia, Australasia and South Asia .*"

We have also removed the reference to transferability on line 500

20) L.334 Table S3. This summary (mean bias) compensates positive and negative bias regions. It is not a performance metric anymore. A value of zero can be achieved by wild, opposite biases. See e.g. WAS-22 T2m JJA. It also penalizes biases of the same sign (EUR44-CCLM4 T2m JJA: 1K) more than opposite biases (EUR44-CCLM5 T2m JJA: 0.38K). It is also not very useful to compare across regions (e.g the wild opposite biases in AFR-44 T2m JJA score 0.27K). I would suggest using the mean absolute bias or a quadratic mean if you'd like an extra

penalty for large biases. Are values masked out in Figure 2 included in the mean bias? It would be helpful to add this measure to the panels of this figure to avoid going back and forth between Fig. 2/3 and Table S3. The table is still OK to summarize other seasons. A background color in the table cells according to the value of the score would also be helpful to easily unveil bias patterns.

We see there was a typo, it should be Table S4, but this is now corrected.

Thanks for pointing out this. We have now replaced the mean bias in table S4 with mean absolute bias, added a background color in the table, and the values are also included in subplots in figures 2 and 3, and S11 and S12. The associated text has been updated accordingly. The values masked out in Figure 2 are included in the calculations of the mean absolute biases, and this is explained in the S4 table caption.

21) L.367-369 This might change when using a score that does not compensate opposite biases

The text is now updated according to the new table.

22) L.428-433 Remind the spectral nudging in this paragraph outlining simulation differences

This is now included.

23) L.467 "During the winter season, there is a warm bias over Northwest India and a cold bias ..." Add "and the Ethiopian highlands" after NW India.

This is included, thanks for the remark.

24) L.472 The spatial variability depends on the domain, and so does the ability of the models and observations to reproduce it. It is misleading to mix in a single Taylor diagram the scores for different regions. The information in Figure 4 is duplicated in Figures 5-8 (L.501). I would suggest keeping just Figures 5-8 with an extra effort to make the ERA-Interim values more outstanding. Also, properly zoomed Taylor diagrams should go to the main manuscript. Current zoomed versions (Figs. S17-20) in the supplementary material do not have proper axes and span different areas of the Taylor diagram. Thus, they can hardly be compared to each other.

See our next reply.

25) Figure 7 shows a good example of my comment above. It seems that observational uncertainty (the spread of symbols corresponding to observational data sets) for DJF precipitation in Africa is smaller than in Europe. This goes against intuition, considering the poor observational coverage of these data sets over Africa. Likely, this is the result of large-scale precipitation gradients in Africa, a domain covering a tropical precipitation belt along with subtropical desert regions. This spatial structure is easily captured by any observational data set (correlation ~0.99) or model simulation (correlation 0.90-0.95). Therefore, different models or

observations for a given domain can be compared in a single Taylor diagram.
However, the comparison across domains (as in Figure 4) is more tricky.

Thanks for these comments, we agree that one cannot expect that the spatial
variability is independent of the location. For that reason, we have normalized the
standard deviation according to each domain's observational average. Yet, we agree
with the reviewer that due to geographical and climatological reasons some aspects
of the spatial variability might be easier or harder to capture for a model depending
on the location on earth. But in a sense this is what we want to show with this figure:
CCLM performs differently depending on the domain and simulation setup.

To address your comment, we have included a sentence on lines 491-497 where we
emphasise these points that you mention related to Figure 4. We do think that
keeping Figure 4 gives an advantage to the reader where it is easy to visually
compare the model results for the different domains. The added text reads:

"*...Note that here we use the ensemble mean over all observational datasets,
whereas in Figure 2 and 3 the spread between the observations is taken into
account. Moreover, it should be stressed that the spatial variability is varying
substantially between the domains, and also the quality of the observations (e.g.
very sparsely
observational coverage in Africa compared to Europe), which is again influenci
ng model performance displayed with the
Taylor diagram in Figure 4. Thus, Figure 4 is merely
meant to facilitate a visual comparison of the model results, which can be
challenging to detect in Figure 2-3. A more detailed investigation of the spatial
variability for each domain separately is given in 4.2.*"

Regarding the zoomed figures in the supplementary section: We have improved the
zoomed parts related to Figure 5 and 6 and included them in the main manuscript.
We feel that Figure 7 and 8 does not need to be zoomed and left them unchanged.
As a result of these changes, figure S17-20 is removed from the supplementary
section.

26) L.530-535 "the choice of the driving data has a bigger influence on the
performance" (also L.632) It is not clear what performance you are referring to. RCM
performance can only be assessed with "perfect" boundary conditions (i.e.
reanalysis, ERA-Interim in this case).

Thanks for pointing this out. To avoid confusion, we have replaced the wording
"performance" with "simulated results" a few places, and included some more
description in the text, where we clarify what we mean with performance and how we
can assess it. See also our next reply.

27) The authors discuss GCM-driven simulations on an equal footing with evaluation
simulations (L.541-542). GCM-driven simulations incorporate errors from the GCM
and RCM. In order to disentangle both error sources, GCM-driven simulations should
be compared to evaluation, reanalysis-driven simulations, not to observations. All
GCM-driven simulations should "perform" worse than evaluation simulations.
Otherwise, an error compensation would be occurring between the GCM and the
RCM (this is not desirable). Also, added value can be discussed with GCM-driven

simulations. The authors mix the added value with the garbage-in/garbage-out problem (L.561-563). A simple added value measure could easily be incorporated in the Figures by adding the biases and/or Taylor diagram points corresponding to the driving GCM output (e.g. as the crosses used for Raw ERA-Interim). Given the simulations considered in the study, a clearer discussion of RCM performance, GCM/RCM errors, garbage-in/out and added value should be provided.

We thank the reviewer for this comment.

We have now reorganized section 4.3, to distinguish a bit better the discussion of RCM performance, GCM/RCM errors, garbage-in/out and added value. Moreover, we want to emphasize that we are not discussing in detail the added value of RCM over the GCMs, as we don´t include the results of the GCMs, but it is a good point to discuss how the GCM-driven results differ from the evaluation runs. We have therefore included on lines 610-621 a description where we point this our (e.g. that the GCM-driven results are typically worse than the performance of the evaluation simulation):

 *"To assess whether there is an added value of the downscaled results compared to the GCMs is beyond the scope of this study, as we are focusing on presenting the RCM results and how they are different depending on various configurations and resolutions. However, it should be noted from Figure 5 - 8 that the performance of the GCM-driven simulations, estimated in the Taylor diagrams, is typically in the same range as for the evaluation simulations for temperature. For precipitation the evaluation simulations generally perform closer to the observations than the GCM-driven simulations. These results indicate that there is no error compensation between the GCMs and the RCMs."*

Regarding including the GCMs in the analysis, we decided to not do so, mainly because to reduce the amount of data, but also because that would be a completely different study, where the focus would not be necessarily to document the model development but rather to assess the GCM-RCM model chains, and we are now stating this on lines 527-529.

The garbage in/garbage out problem is not intended to be discussed in detail in this study (briefly mentioned on line 610-615), but our main goal with including the discussion of the GCM-driven simulations was that we wanted to highlight how much (or how little) influence the RCM with different configurations and model resolutions can have on the downscaled results. Before we started to describe the results (section 4 on lines 334-336), we had described the purpose of analyzing the evaluation simulations and the GCM driven simulations, but we now add some text to emphasise this as follows:

*"... In the next step, the results of the GCM-driven historical simulations (1981-2010, whereby RCP85 is used for 2006-2010) are analysed, whereby we extend the discussion to include the choice of forcing data and the effect of various model configurations and resolutions."*

Moreover, we point the reviewer to the summary and outlook section at lines 676-681, where we state that the model results have a dependency on the driving data, which is larger than the dependency on the various model resolutions and

configurations, seen in particular for the precipitation. We do not say anything about added value, beyond that you only change your model results slightly with changing the resolution or configuration, but if you change your driving GCM, you can get very different model results. We think that is one of the main findings from this study, and feel that what we already have written is explaining this.

Overall, we have gone through the manuscript to clarify and change the text when we discuss the downscaled model results, with the aim to avoid to use model performance in when we are describing the downscaled GCM results, and also modified some of the text in section 4.3 and 5 to clarify in the wording regarding RCM performance and added value.

27) L.544 "the bias patterns and model performance are" Rephrase. Bias is also a model performance measure

We have now changed it to (on line 574) "*...how the model results and performance are influenced by changing the model configuration, model version, horizontal resolution, ..*"

28) L.545-547 "We have shown [...] that the model also has to be re-tuned to obtain a model configuration that is optimal for the domain" Well, strictly, you have only shown re-tuned results for each domain. The need for that has been left to previous references.

Good point, we have rephrased it to (on lines 576-579):

*" An added value in terms of model performance is not necessarily gained by sloley increasing the horizontal resolution, but to change the model configuration that is optimal for the domain has advantages. This study is meant to document these re-tuning experiences that can be used when designing future CORDEX simulations. "*

Also, L.635 "The results from this large COSMO-CLM model ensemble indicate that an RCM-modeler can do a lot when it comes to improve the model performance" This is not derived from your results. You can start the sentence at "An RCM-modeler ..."

Changed.

29) L.633-635 Please, rephrase for better readability. Split into two simpler sentences?

Changed to: "*...the model results have a dependency on the driving data, seen in particular for the precipitation. When changing the resolution or slightly altering the model configuration, the simulated results are only marginally modified. However, if a substantial adjustment is done in the model configuration (such as coupling to a different land model as done for AUS-44),  the model results differ more.*"

30) L.647 Please, expand or clarify what ICON is.

included.

31) Small typos:

L.354 micorphysics

Changed.

L.373 remove "As" at the beginning of the sentence

Changed.

L.387 remove parenthesis around (Panitz et al, 2014)

Changed.

L.522 more close -> closer

Changed.

L.595 2018 -> 2020

Changed.

L.679 publically -> publicly

Changed.

---

## Author Comment (AC3)

**Response to Anonymous Referee Comment (RC2), 16 May 2021
([https://doi.org/10.5194/gmd-2020-443-RC2](https://doi.org/10.5194/gmd-2020-443-RC2) ):**

We much appreciate the positive assessment provided by the reviewer.
All parts in the manuscript pointed out by the reviewer have been adapted or
clarified.  A point-by-point response to the comments from the reviewer is given
below, where for convenience we have copied in the reviewer's comments followed
by our reply (the reviewers' comments are in blue text). When we are referring to a
specific line, if nothing else is stated, we are using the line numbering in the
document with the track change.

A Review of "COSMO-CLM Regional Climate Simulations in the CORDEX
framework: a review" by Sørland et al.

General comments

This is a review paper intended to document the development, progress, and
performances of COSMO-CLM regional climate simulations over a number of
CORDEX domains including Europe, Africa, South Asia, East Asia and Australasia.
While there have been many scientific publications of COSMO-CLM over individual
domains, this review paper inter-compares its performances among different
domains. I find this paper is useful and can provide guidance to future work in
regional climate simulations especially in non-native domains of COSMO- CLM. The
structure and logical flow of this review paper made it easy to read and follow. I have
a few suggestions for further improvement.

Specific comments

1. In Figure 5 to 8, the last Taylor Diagram was labelled as "South-west Asia". I
suppose this refers to "South Asia". Perhaps should be changes should be made to
supplementary figures as well.

Changed

2. "East Asia" was one of the CORDEX domains included in this study. However, this
"East Asia" is the "old" domain prior to the establishment of "Southeast Asia"
CORDEX domain (e.g. Tangang et al. 2020; www.cordex.org). The new "East Asia"
CORDEX domain was shrunk a smaller domain. Hence, a sentence is needed to
explain this and avoid confusion.

We had already included this information in the text and have now added more
details (marked in italics) on lines 443-447 to make the reader better aware of this:

"CORDEX simulations over East Asia at 0.44 (EAS-44) and 0.22 (EAS-22) have
been performed with version CCLM5-0-2 and CCLM5-0-9, respectively. Due to an
updated EAS-CORDEX domain, the domains are not identical: while the EAS-44 is
following the CORDEX framework for the first phase *which covers a large area*

*including southeast Asia and northern Australia*, the EAS-22 is following the second phase *with a smaller domain excluding tropical southeast Asia* (Zhou et al., 2016). *Note that a Southeast Asia CORDEX domain has been established (Tangang et al. 2020).*"

.

3. While much have been written on the performances of COSMO-CLM among its different versions and regions, very little review was provided on how this RCM fares compared to other RCMs in different regions.

In Figure 1, we compare the COSMO-CLM performance over Europe to the other RCMs. It would be ideal to include other RCMs for the other regions as well, but that needs to be a study of its own. We do refer to studies we are aware of, that are comparing different RCMs for the different CORDEX domains, see e.g. on line 209 ("... and compared to the other CORDEX-Africa RCMs in a number of studies (e.g., Dosio et al., 2019, 2020).") . and on line 227 ("The CCLM4-8-17-CLM3-5 simulations are analyzed in model comparison studies (Di Virgilio et al., 2019; Hirsch et al., 2019) over the Australian part of the CORDEX-Australasia domain."

4. List of GCMs in lines 295 – 303 is better placed in a proper table.

We have now included the GCMs in a table (Table 2).

5. I am not sure the real purpose of having a detailed analysis of evaluation of GCM driven simulations in the context of providing a review on COSMO-CLM here? In Figure 5 to 8, the performances of various GCM driven runs were shown to be different. However, these differences are expected and reflect inter-GCM differences. I don't see the relevance of this detailed analysis here in reviewing COSMO-CLM performances.

This is a valid point, and the reviewer is correct that the results from the different downscaled GCMs are expected to diverge. However, one of the motivations for including these simulations in this study is to explore how much the results from e.g. one downscaled GCM, vary when there are changes in the model configuration or horizontal resolution. For instance for temperature, changing the model resolution or configuration (marginally) is not influencing the results a lot, but for precipitation,  the model resolution and configuration plays a bigger role (see lines 557-561).

We have also made some changes in the manuscript (particularly regarding section 4.3) related to distinguishing the discussion of the RCM performance, GCM/RCM errors, and added value. See also the reply to RC1 (comment 26).

6. I think in the "Summary and outlook" section, the authors did not adequately address the issue of uncertainty in the simulations. Even within different versions of COSMO-CLM, we can see different biases (e.g. Figure 2 and 3). What does this mean in terms of uncertainty of using COSMO-CLM in different regions? Should this review paper recommend the use of different RCMs i.e. multi-RCM approach?

We cannot address the uncertainty with only one RCM, for this it is needed to span the full 3-D GCMxRCMxRCP (or 4-D matrix if you include all the realizations), thus

this review paper only explores a subsample of the uncertainty. We never state that we will address the uncertainty in regional climate projections in this study, but  this review paper is more meant to guide RCM modelers (and also GCM modelers) in how to design their experimental settings, which is particularly important now before next CORDEX simulations will be started. We state the scope of the manuscript on line 91-97 in the introduction as: *"... in this study we assess and compare the model performance over Europe with the four CORDEX-CORE domains Africa, East Asia, Australasia and South Asia. Since the existing COSMO-CLM CORDEX simulations differ in more than one way (i.e. versions, configurations and resolutions), we do not perform a systematic analysis of each simulation but rather a qualified judgement, based on all model integrations that are currently available (as of February 2020). Such an analysis will support the future design of model simulations in the community. The dependence of the model results on the driving GCM is also discussed."*

and  in the summary and outlook on lines 634-636, we summarize the main results with respect to the scope of the study (see e.g. *"Nevertheless, our analysis of all the available model runs, can provide guidance for the future design of regional climate projections by the CLM-Community as well as by other RCM-groups. Moreover, as the focus on downscaling CMIP5 GCMs will be replaced by CMIP6 in the near future, we anticipate this is a good time to reflect how coordinated RCM simulations can contribute in an optimal way."*)

Ref:

Tangang et al. 2020. Projected Future Changes in Rainfall in Southeast Asia based on CORDEX – SEA Multi-model Simulations. Climate Dynamics, 55, pages1247–1267, https://doi.org/10.1007/s00382-020-05322-2

Thanks for the suggested reference, which is now included on line 447